# Rescue of deficits by *Brwd1* copy number restoration in the Ts65Dn mouse model of Down syndrome

Sasha L. Fulton [1,20], Wendy Wenderski[2,3,4,5,20], Ashley E. Lepack[1,20], Andrew L. Eagle [6], Tomas Fanutza[7], Ryan M. Bastle[1], Aarthi Ramakrishnan[1], Emma C. Hays [1], Arianna Neal[1], Jaroslav Bendl [8,9,10], Lorna A. Farrelly[1], Amni Al-Kachak[1], Yang Lyu[1], Bulent Cetin[1], Jennifer C. Chan[1], Tina N. Tran[11,12], Rachael L. Neve[13], Randall J. Roper[14], Kristen J. Brennand [1,8,9,15,19], Panos Roussos [8,9,10,16], John C. Schimenti [11,12], Allyson K. Friedman [17], Li Shen [1], Robert D. Blitzer [7,8], Alfred J. Robison[6], Gerald R. Crabtree [2,3,4,5] & Ian Maze [1,18] ✉

With an incidence of ~1 in 800 births, Down syndrome (DS) is the most common chromosomal condition linked to intellectual disability worldwide. While the genetic basis of DS has been identified as a triplication of chromosome 21 (HSA21), the genes encoded from HSA21 that directly contribute to cognitive deficits remain incompletely understood. Here, we found that the HSA21-encoded chromatin effector, *BRWD1*, was upregulated in neurons derived from iPS cells from an individual with Down syndrome and brain of trisomic mice. We showed that selective copy number restoration of *Brwd1* in trisomic animals rescued deficits in hippocampal LTP, cognition and gene expression. We demonstrated that Brwd1 tightly binds the BAF chromatin remodeling complex, and that increased *Brwd1* expression promotes BAF genomic mistargeting. Importantly, *Brwd1* renormalization rescued aberrant BAF localization, along with associated changes in chromatin accessibility and gene expression. These findings establish BRWD1 as a key epigenomic mediator of normal neurodevelopment and an important contributor to DS-related phenotypes.

Down syndrome (DS) is the most common form of autosomal aneuploidy in humans, and is characterized by physical growth delays, skeletal abnormalities, neurological deficits and cognitive impairments[1]. Although the genetic cause of DS is full or partial triplication of chromosome 21 (HSA21)[2,3], triplicated genes on HSA21 do not fully account for the widespread transcriptional dysregulation observed in DS. Recent RNA-sequencing of human postmortem DS brain tissues demonstrated robust gene expression changes across all chromosomes (both up- and downregulated) throughout development[4], with ~70% of triplicated HSA21 genes being subject to

dosage compensation, which buffers against increased expression[5]. Numerous studies in both human-derived cells, brain tissues[4,6], and rodent models of DS[1,7,8] have found that there is not a clear 1:1 relationship between gene dosage and gene expression at trisomic loci, further suggesting that DS phenotypes may be driven by more complex regulatory mechanisms. In addition, given the tremendous amount of variability in the severity and clinical presentation of DS[5], it is believed that epigenetic processes may also contribute to global patterns of transcriptional dysregulation, both during neurodevelopment and in adulthood. However, our understanding of how

chromatin-based mechanisms contribute to DS phenotypes remains limited. HSA21 encodes several chromatin regulators[9], including BRWD1[10], a WD-repeat and bromodomain-containing protein[10]. In previous studies focusing on germ[11] and immune cells[12,13], BRWD1 was shown to modulate chromatin structure via proposed interactions with the mammalian SWI/SNF (BAF) ATP-dependent chromatin-remodeling complex[14,15]. However, despite being encoded on HSA21, and its demonstrated activity as an epigenetic regulator, potential roles for BRWD1 in the context of HSA21 triplication and DS patho-physiology have not yet been explored.

Here, we found that BRWD1 is upregulated in both trisomic human neurons and Ts65Dn mouse brain. Selective restoration of *Brwd1* copy number in trisomic mice rescued DS-related cognitive impairments, neuronal physiology, and alterations in transcription. In adult euploid mice, acute Brwd1 overexpression in dorsal hippocampus was sufficient to impair memory, attenuate activity-induced gene expression and promote excitation/inhibition imbalance. Furthermore, we found that Brwd1 tightly associates with the BAF complex in both embryonic and adult brain, and that restoring *Brwd1* copy number in trisomic animals substantially rescued alterations in BAF genomic localization, as well as associated changes in neuronal chromatin accessibility. These data demonstrate a dosage-sensitive role for Brwd1 in targeting BAF complexes to appropriate loci within the central nervous system, and indicate a central role for BRWD1 in the precipitation of neurological deficits associated with DS.

## Results

### BRWD1 is upregulated in neurons derived from iPS cells from an individual with DS and DS-like rodent brain

Given that not all HSA21 genes are dysregulated at the level of transcription in DS, we first generated human induced pluripotent stem (iPS) cell-derived forebrain neurons from a DS subject to assess whether *BRWD1* levels are indeed increased as a result of HSA21 triplication. In doing so, we found that *BRWD1* expression was elevated in neurons derived from iPS cells from an individual with DS *vs.* a respective isogenic control (Supplementary Fig. 1a, b). Next, in order to explore potential mechanistic roles for BRWD1 in mediating trisomy 21-related phenotypes, we measured *Brwd1* expression in brain tissues of Ts65Dn mice, a well characterized model of DS with segmental trisomy 16 and a corresponding copy number triplication for approximately half of the homologous HSA21 genes, including *BRWD1*[16–18]. Consistent with previous studies in humans with DS and trisomic mice[1,4,18,19], we found that *Brwd1* mRNA was significantly elevated in both embryonic (E17.5) and adult (6-week) male and female Ts65Dn brain tissues, including the prefrontal cortex (PFC), hippocampus and cerebellum—with no significant difference between the sexes in adult euploid animals (Fig. 1a and Supplementary Fig. 2a, b).

To further confirm the etiological relevance of this trisomic mouse model to human DS at the level of transcription, we next performed RNA-seq profiling on E17.5 Ts65Dn *vs.* euploid forebrain tissues. Differential expression analyses demonstrated that the Ts65Dn mouse model exhibits robust transcriptional changes that overlap significantly with human DS single-nuclei gene expression profiles in postmortem PFC (Supplementary Fig. 3a, b)[20]. Of note, we found that only upregulated genes in Ts65Dn *vs.* euploid mice overlapped significantly with human DS-associated gene expression, which included genes enriched for pathways associated with cellular development, neuronal differentiation, and synaptic transmission. These data suggest that inappropriate induction of transcripts related to neuronal function may contribute to DS-related phenotypes in Ts65Dn mice (Supplementary Fig. 3c).

### BRWD1 copy number restoration rescues physiological and cognitive hippocampal deficits in adult male Ts65Dn mice

Given that *Brwd1* expression was found to be upregulated in Ts65Dn mouse brain, we next examined whether Brwd1 itself may contribute

to DS-related phenotypes by genetically restoring *Brwd1* copy number in otherwise trisomic mice. We crossed Ts65Dn females to hetero-zygote *Brwd1*[+/− 11] males, resulting in male and female offspring of four genotypes: euploid, *Brwd1*[+/−], Ts65Dn, and Ts65Dn;*Brwd1*[+/−] (Fig. 1b). Importantly, the Ts65Dn;*Brwd1*[+/−] cross selectively rescued *Brwd1* tri-plication observed in male Ts65Dn mice without directly genetically restoring the trisomic background (Fig. 1c). However, in female mice, while *Brwd1* expression was significantly reduced in Ts65Dn; *Brwd1*[+/−] *vs.* Ts65Dn animals, we identified only trending *Brwd1* increases in Ts65Dn *vs.* euploid mice when comparing all four genotypes (Supplementary Fig. 2c), reflecting higher variability in *Brwd1* upregulation in female Ts65Dn hippocampus. Indeed, female Ts65Dn mice displayed more modest hippocampal *Brwd1* fold-change (FC) differences *vs.* euploid (average 1.19 FC) in comparison to those detected in males (average 1.41 FC). These findings are consistent with previous studies in which BRWD1 has been shown to display sex-specific functions and expression levels, particularly in the context of early reproductive cellular genesis[11,21].

Next, to examine the impact of *Brwd1* copy number restoration on hippocampal synaptic function in adult mice (6-week), we performed ex-vivo brain slice long-term potentiation (LTP) recordings, where LTP was induced by theta-burst stimulation at Schaffer collateral inputs to area CA1. Consistent with previous studies[22,23], LTP was significantly impaired in slices from Ts65Dn male mice vs. euploid controls - however, this deficit was fully rescued in Ts65Dn;*Brwd1*[+/−] animals, while *Brwd1* heterozygosity did not affect LTP (Fig. 1d). These deficits were specific to more stable forms of neuronal plasticity, as other measures of baseline synaptic function were unaffected in Ts65Dn animals (Supplementary Fig. 4a, b). Interestingly, in females, LTP was not affected in Ts65Dn mice *vs.* euploid controls, and no differences in LTP were observed comparing euploid females *vs.* Ts65Dn; *Brwd1*[+/−] or *Brwd1*[+/−] genotypes (Supplementary Fig. 2d). These findings parallel previous reports that have identified regional differences in brain function between male and female Ts65Dn mice, as well as in other mouse models of DS[24,25], suggesting that males may exhibit more severe hippocampal deficits *vs.* females.

Next, to investigate whether hippocampal-dependent cognitive deficits associated with DS might be improved in Ts65Dn;*Brwd+/-* animals, we employed a rodent contextual fear conditioning (FC) paradigm to assess learning and memory in the four genotypes described above. Male Ts65Dn mice displayed a significant reduction in FC memory *vs.* euploid controls—a deficit that was fully rescued with *Brwd1* renormalization, with no effect of *Brwd1* heterozygosity observed (Fig. 1e). Although we did not detect altered hippocampal LTP in female Ts65Dn mice, we did find that female Ts65Dn mice scored worse in the contextual FC memory task *vs.* euploid controls (Supplementary Fig. 2e), suggesting that distinct molecular mechanisms may contribute to contextual fear learning in male vs. female Ts65Dn mice. Furthermore, *Brwd1* copy number restoration did not significantly rescue these cognitive deficits, reflecting a more limited contribution of Brwd1 to Ts65Dn hippocampal function in females. Together, these data indicate that in male animals, increased *Brwd1* gene dosage is necessary for the precipitation of DS-related physio-logical and cognitive hippocampal deficits in the Ts65Dn mouse model.

### BRWD1 overexpression in adult male hippocampus is sufficient to induce cognitive and physiological defects in euploid animals

We then investigated whether acute *Brwd1* overexpression in euploid animals may be sufficient to cause DS-related cognitive deficits. We generated herpes simplex virus (HSV) vectors to overexpress BRWD1 (HSV-BRWD1-GFP) *vs.* an empty vector control (HSV-GFP) in adult male dorsal hippocampus (CA1), followed by a battery of behavioral assays to assess the impact of *Brwd1* overexpression on context- and spatial-

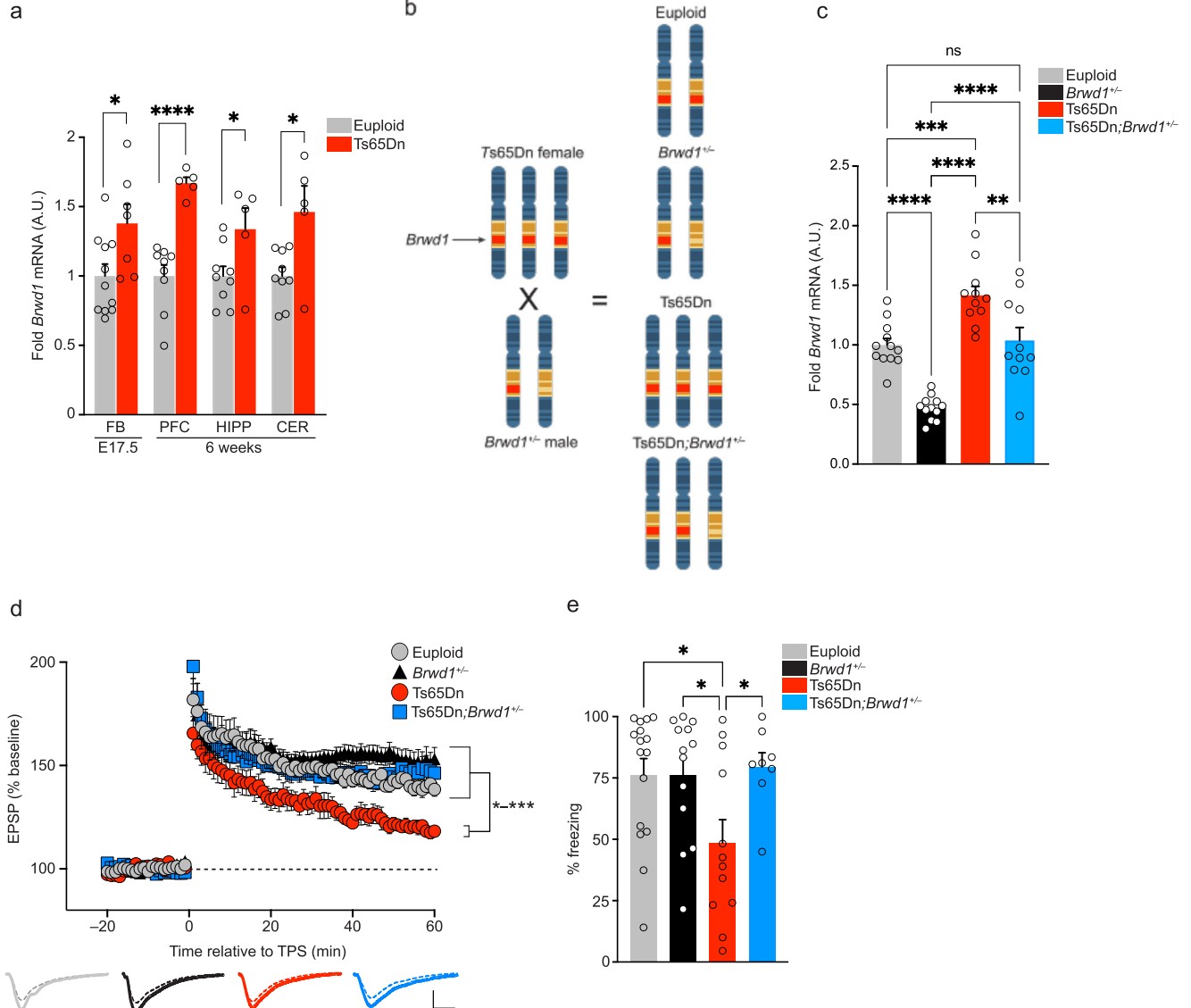

**Fig. 1 | *Brwd1* copy number restoration rescues synaptic and cognitive deficits in male trisomic animals. a** qPCR expression data for *Brwd1* in embryonic day (E) 17.5 forebrain (FB) and adult (6-week) PFC, hippocampus (HIPP) and cerebellum (CER) from euploid *vs.* Ts65Dn male mice. A.U. = Arbitrary Units, with experimental (non-euploid) group averages normalized to respective euploid controls. **b** Schematic depicting the generation of mouse genotypes to be investigated. **c** qPCR expression data for *Brwd1* in euploid *vs. Brwd1*[+/−] vs. Ts65Dn vs. Ts65Dn;*Brwd1*[+/−] E17.5 forebrain. **d** Deficiency of hippocampal LTP in adult male (6-

week) Ts65Dn mice is rescued by *Brwd1* copy number restoration. The representative traces were recorded at the end of the baseline period (dashed lines) and 60 min after induction of LTP (solid lines). Calibrations: 0.5 mV/5 ms. **e** Context dependent fear conditioning—displayed as % freezing in the trained context—comparing euploid *vs. Brwd1*[+/−] vs. Ts65Dn *vs.* Ts65Dn;*Brwd1*[+/−] mice. Data are presented as averages ± SEM. See Supplementary Information Materials for full caption with *n*'s and statistics. Source data are provided as a source data file.

dependent learning and memory, as well as anxiety and anhedonia related behaviors. We found that selectively upregulating *Brwd1* expression to levels similar to those observed in Ts65Dn animals was sufficient to elicit significant cognitive deficits in hippocampal-dependent tasks (Supplementary Fig. 5a–o). Next, to explore the electrophysiological impact of overexpressing *Brwd1* in adult euploid male hippocampus, we performed patch clamp recordings of virally infected (HSV-BRWD1-GFP *vs.* HSV-GFP) CA1 pyramidal neurons. Our results indicated that *Brwd1* overexpression significantly increased both neuronal excitability (Supplementary Fig. 6a) and the frequency of spontaneous excitatory post-synaptic currents (sEPSC; Supplementary Fig. 6b), while significantly reducing the frequency of inhibitory post-synaptic currents (sIPSC; Supplementary Fig. 6c). sEPSC and sIPSC amplitudes were unaffected by *Brwd1* overexpression (Supplementary Fig. 6b, c). Finally, we observed reduced immediate early gene

(IEG) induction in *Brwd1* overexpressing CA1 following exposure to a novel environment (Supplementary Fig. 6d), suggesting that BRWD1 overexpression leads to shifts in excitation/inhibition balance and neuronal plasticity that may prevent stable induction of normal LTP. Together with our previous results, these data demonstrate that *Brwd1* upregulation in DS-like brain is both necessary and sufficient to promote synaptic and cognitive deficits associated with the disorder, and that *Brwd1* copy number restoration fully rescues these effects in male mice.

## BRWD1 triplication contributes to gene expression abnormalities in DS-like brain

We next sought to explore Brwd1's role in regulating neuronal gene expression patterns in DS-like brain by performing RNA-seq analyses on primary neuronal cultures derived from euploid *vs.* Ts65Dn *vs.*

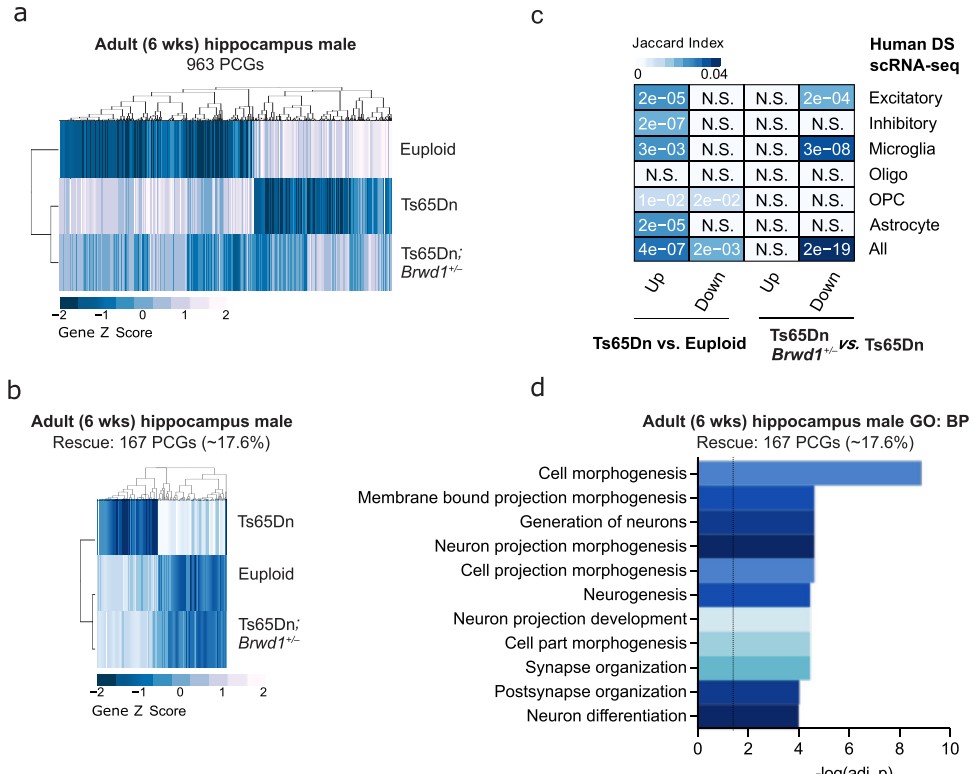

**Fig. 2 | Rescue of aberrant gene expression in male trisomic brain by *Brwd1* renormalization. a** RNA-seq heatmaps of DE genes comparing euploid vs. Ts65Dn vs. Ts65Dn;*Brwd1*+/− adult male (6-week) hippocampus. Normalized RNA expression values (averaged between replicates) were used to generate z-scores for each row. **b** Normalized heatmaps of RNA expression values for DE genes in adult male hippocampus that display pairwise significant regulation between Ts65Dn vs. euploid, and are rescued in Ts65Dn;*Brwd1*+/− vs. Ts65Dn. **c** Heatmap displays Jaccard index, as well as adjusted p-values, from odds ratio analyses of the overlap between DE genes from euploid vs. Ts65Dn vs. Ts65Dn;*Brwd1*+/− comparisons and previously published human DS single-nuclei RNA-seq data *vs.* age-matched controls[20]. **d** Bar graph of −log10(adj. p-val) for gene ontology (GO) processes displaying enrichment for PCGs identified in **c** above. See Supplementary Information Materials for full caption with *n*'s and statistics. Source data are provided as a source data file.

Ts65Dn;*Brwd1*+/ embryonic mouse forebrain (mixed male and female). Using this embryonic neuronal system, which was maintained in the absence of proliferating glial cells, we observed widespread changes in gene expression, detecting 9588 differentially expressed (DE) protein coding genes (PCGs) when comparing across all three genotypes in a likelihood ratio test (Supplementary Fig. 7a). Separate pairwise comparisons between Ts65Dn vs. euploid and Ts65Dn;*Brwd1*+/− vs. Ts65Dn groups revealed that the majority of differentially expressed PCGs in Ts65Dn vs. euploid neurons were reversed (~60.3% rescue) by *Brwd1* copy number restoration (Supplementary Fig. 7b), with rescued genes significantly enriched for gene ontology (GO) terms including axogenesis, synaptic transmission and regulation of neuronal morphology (Supplementary Fig. 7c). These data demonstrate that Brwd1 plays a critical role in maintaining normal patterns of neuronal gene expression during brain development.

In order to further evaluate the effects of Brwd1 rescue within the context of adult brain, we performed RNA-seq on hippocampal tissues from 6-week-old male animals, comparing Ts65Dn *vs.* euploid and Ts65Dn;*Brwd1*+/− genotypes (Fig. 2a). Pairwise comparisons identified 963 DE genes (FDR < 0.1) between Ts65Dn vs. euploid, and 801 DE genes (FDR < 0.1) between Ts65Dn;*Brwd1*+/− vs. Ts65Dn, with ~17% of DE genes in Ts65Dn vs. euploid mice being significantly reversed in their expression with *Brwd1* copy number restoration (Fig. 2b). Consistent with our earlier analyses examining E17.5 forebrain tissues (Supplementary Fig. 3), odds-ratio assessments revealed significant overlaps with human DS associated gene expression, specifically for PCGs upregulated in Ts65Dn vs. euploid, and for those downregulated in Ts65Dn;*Brwd1*+/− vs. Ts65Dn animals, indicating that *Brwd1* copy number restoration significantly reverses trisomic gene expression

patterns in Ts65Dn male hippocampus that are relevant to human DS (Fig. 2c). Functional annotation analysis of rescued PCGs in adult male hippocampus against GO databases demonstrated significant enrichment of gene sets related to neuronal differentiation, neuronal morphology and synaptic function, consistent with observed deficits in synaptic function and hippocampal memory in male Ts65Dn mice (Fig. 2d).

In female mice, a more modest rescue (~9.2%) of hippocampal gene expression changes with *Brwd1* normalization was observed (Supplementary Fig. 8a, b). Associated processes/pathways for rescued genes in female mice were distinct from those seen in males, with significant GO term enrichment identified for protein synthesis and translation, as well as neuronal differentiation (Supplementary Fig. 8c). In addition, the genes that were found to be differentially expressed between female vs. male Ts65Dn most significantly associated with LTP and neuronal morphology, highlighting potential molecular and/or anatomical differences between the sexes in the pathophysiology of DS-related deficits (Supplementary Fig. 8d). Notably, in all cases (E17.5 forebrain, adult male and female hippocampus), differentially expressed genes from Ts65Dn animals were not limited to trisomic loci, consistent with previous findings[4,26], and *Brwd1* renormalization reversed the expression of many of these genes across all chromosomes (Supplementary Fig. 9a).

**Brwd1 interacts with the BAF chromatin remodeling complex to influence its genomic targeting in DS-like brain**

To further explore the biochemical mechanisms through which BRWD1 exerts its effects on synaptic plasticity, cognition and gene expression in DS, we sought to examine whether BRWD1 might

function independently or as a part of a protein complex. Given the current lack of validated antibodies for BRWD1 in rodent brain tissues, we generated and fully validated knock-in mice that endogenously express BRWD1 with C-terminal FLAG-HA tags (Brwd1[FLAG-HA]) (Supplementary Fig. 10a–g). We precipitated soluble nuclear proteins from E17.5 or adult, wildtype (untagged) vs. Brwd1[FLAG-HA] whole mouse brains with ammonium sulfate (Fig. 3a) and detected specific HA signal at the expected size of endogenous BRWD1-FLAG-HA protein by Western blot (Supplementary Fig. 11a, b). Native soluble nuclear proteins were then separated by density over a 10–30% glycerol gradient. The gradient fractions were immunoblotted for HA to indicate endogenous BRWD1-FLAG-HA and for nuclear protein complexes of known size. A single peak of full-length endogenous BRWD1-FLAG-HA was detected in high-density fractions containing the canonical BAF (~2 MDa) and PBAF (~3 MDa)[27] complexes (Fig. 3b and Supplementary Fig. 11c). Notably, BRWD1-FLAG-HA (261 kDa) was not detected in low-density fractions containing AP-1 (160–440 kDa)[28]. These results suggested that BRWD1 functions as part of a large protein complex.

Given BRWD1's role as a chromatin interacting protein[12–14], we next tested the hypothesis that BRWD1 directly interacts with the BAF complex by performing a series of BAF immunoprecipitation (IP) experiments from embryonic or adult, wildtype vs. Brwd1[FLAG-HA] brain nuclear extracts. At E17.5, the embryonic mouse brain is predominantly composed of immature neurons and neural progenitors[27]. Upon mitotic exit, neural progenitor BAF (npBAF) subunits SS18, ACTL6A (BAF53a) and DPF2 (BAF45d) are downregulated, and neuronal subunits SS18L1 (CREST), ACTL6B (BAF53b) and DPF1/3 (BAF45b/c) are upregulated, leading to the formation of the neuronal-specific BAF (nBAF) complex[27,29–31]. To learn if BRWD1 associates with BAF complexes during neural development, we immunoprecipitated (IP'd) BAF complexes from E17.5 brain nuclear extracts with antibodies to the homologous subunits SS18 (npBAF) or SS18L1 (nBAF), to the core BAF ATPase SMARCA4 (BRG1) or to IgG as a control. Endogenous BRWD1-FLAG-HA robustly co-IP'd with antibodies to SMARCA4 and SS18 (npBAF), and to a lesser extent SS18L1 (nBAF), indicating that BRWD1 associates with both npBAF and nBAF complexes in embryonic brain (Fig. 3c). We next immunoprecipitated BAF complexes from adult wildtype vs. Brwd1[FLAG-HA] brain. We found that BRWD1-FLAG-HA co-IP'd with antibodies to core BAF subunits SMARCA4, SMARCC2 and SMARCB1; and with two different antibodies to the neuronal subunit SS18L1 (Fig. 3d and Supplementary Fig. 11b). Next, the stability of the BAF:BRWD1-FLAG-HA interaction was challenged with increasing concentrations of the denaturing agent, urea. In previous studies, transcription factor interactions with BAF were destabilized in as little as 0.25 M urea, while core BAF subunits resisted denaturation in up to 4 M urea[32,33]. Although ~60% of BRWD1 was destabilized in 0.5 M urea, we found that 40% remained bound to BAF through 4 M urea, surpassing the stability of the dedicated BAF subunit, SMARCB1 (Fig. 3e, and quantified in Fig. 3f). Thus, BRWD1 stably associates with BAF complexes in both embryonic and adult brain.

Since BRWD1 tightly interacted with BAF and co-migrated with it on the gradient, we next explored the possibility that BRWD1 is a dedicated subunit of BAF. Deleting BAF subunits can affect its assembly and migration glycerol gradients[34]. Therefore, we subjected nuclear extracts from Brwd1 knockout mice to density sedimentation on a 10–30% glycerol gradient. Brwd1[-/-] BAF complexes migrated normally, indicating that BRWD1 was not necessary for BAF assembly and did not contribute to the apparent molecular weight of BAF, possibly owing to substoichiometry (Supplementary Fig. 11d). We then subjected brain nuclear extracts from E17.5 or adult Brwd1[FLAG-HA] mice to 3 rounds of immunodepletion with antibodies to BAF subunits or IgG, as indicated in the co-IPs in Fig. 3c, d. Antibodies to SMARCA4, SS18 and SS18L1 selectively and near-completely immunodepleted their target proteins from E17.5 brain nuclear extracts; however, non-target BAF subunits were co-depleted by only 20–30%, and BRWD1-FLAG-HA was

depleted by ~20% only with the SS18 antibody (Supplementary Fig. 12a, b). From adult brain nuclear extracts, antibodies to SMARCA4 and SS18L1 immunodepleted their target proteins by 50–70%, while non-target BAF subunits were depleted by 20–50%, and BRWD1 was depleted by ~20% on average (Supplementary Fig. 12c–e). Although BRWD1 was immunodepleted in a similar manner to BAF subunits, we were unable to fully immunodeplete BAF complexes from nuclear extracts and therefore could not determine if BRWD1 is indeed dedicated to BAF.

Given BRWD1's chromatin binding domains (bromo- and WD40 repeats) and stable interaction with BAF, we next investigated whether Brwd1 upregulation in Ts65Dn mice may affect BAF genomic targeting and whether Brwd1 copy number restoration might rescue any deficits observed. We performed ChIP-seq for the BAF complex using an antibody that recognizes SMARCA2/4 in euploid vs. Ts65Dn vs. Ts65Dn;Brwd1[+/-] hippocampus from adult (6-week) males. In euploid animals, we detected the BAF complex primarily enriched at gene promoters, gene bodies and enhancers (Fig. 4a and Supplementary Fig. 13a–c), loci also marked by active histone PTMs, such as H3K27ac, H3K4me1, H3K4me3, H3K36me3, and H4K20me1) (Supplementary Fig. 13b). Importantly, differential analyses revealed that in Ts65Dn vs. euploid hippocampus, ~60.5% (2156/3563) of all SMARCA2/4 enriched protein coding gene (PCG) sites in euploid animals are altered in trisomic brain, with ~27.6% (595/2,156) of those differentially enriched sites being rescued by Brwd1 copy number restoration (Fig. 4b and Supplementary Fig. 13c). More specifically, in Ts65Dn, we found that SMARCA2/4 binding was lost at promoters (64.1% downregulated events) compared to euploid (Supplementary Fig. 13c), and robustly increased in intergenic regions (90.2% upregulated events) (Supplementary Fig. 13c). Mapping these sites of differential SMARCA2/4 enrichment in mouse Ts65Dn brain against chromatin states identified from human brain, we similarly found that the BAF complex is mistargeted in trisomic hippocampus away from promoters or "poised" promoters and towards repressed genomic regions (e.g., repressed enhancers, heterochromatic loci and repetitive regions of the genome) (Fig. 4c and Supplementary Fig. 14a, b). Further odds ratio analyses revealed that PCGs that lost SMARCA2/4 binding in Ts65Dn vs. euploid hippocampus overlapped most significantly with upregulated genes in both Ts65Dn and human DS, as well as with rescued downregulated genes in Ts65Dn;Brwd1[+/-], suggesting that BAF complex mistargeting away from promoters is associated with inappropriate induction of gene expression (Supplementary Fig. 13d, e). Finally, gene pathways significantly associated with rescued SMARCA2/4-bound PCGs (595 genes from Fig. 4b, c) included neuronal morphogenesis, synaptic transmission, and LTP (Fig. 4d), which is consistent with our earlier findings from bulk RNA-seq analyses presented in Fig. 2.

Based on BAF's chromatin remodeling activity[35], we predicted that sites of differential SMARCA2/4 enrichment may also be associated with changes in neuronal chromatin accessibility. To test this, we performed neuronal-specific ATAC-seq in euploid vs. Ts65Dn vs. Ts65Dn;Brwd1[+/-] hippocampus from adult (6-week) male mice to identify differentially accessible chromatin regions across genotypes (Fig. 4e, Supplementary Fig. 13c and f,g). Consistent with our previous data, we found that Brwd1 copy number normalization significantly restored chromatin accessibility changes observed in Ts65Dn brain, with ~65.1% (5,743/8,827) of differentially accessible PCG loci being rescued in the Ts65Dn;Brwd1[+/-] genotype (Supplementary Fig. 13c). We also detected a highly significant overlap of PCGs that were found to be up-and down-regulated in Ts65Dn vs. euploid and Ts65Dn;Brwd1[+/-] vs. Ts65Dn, respectively (Fig. 4e). Finally, we employed odds ratio analyses to compare PCGs with differential SMARCA2/4 enrichment to PCGs with altered chromatin accessibility in Ts65Dn animals. In doing so, we identified a significant association (~85%, 506/595) between BRWD1-rescued PCGs in both SMARCA2/4 enrichment and chromatin

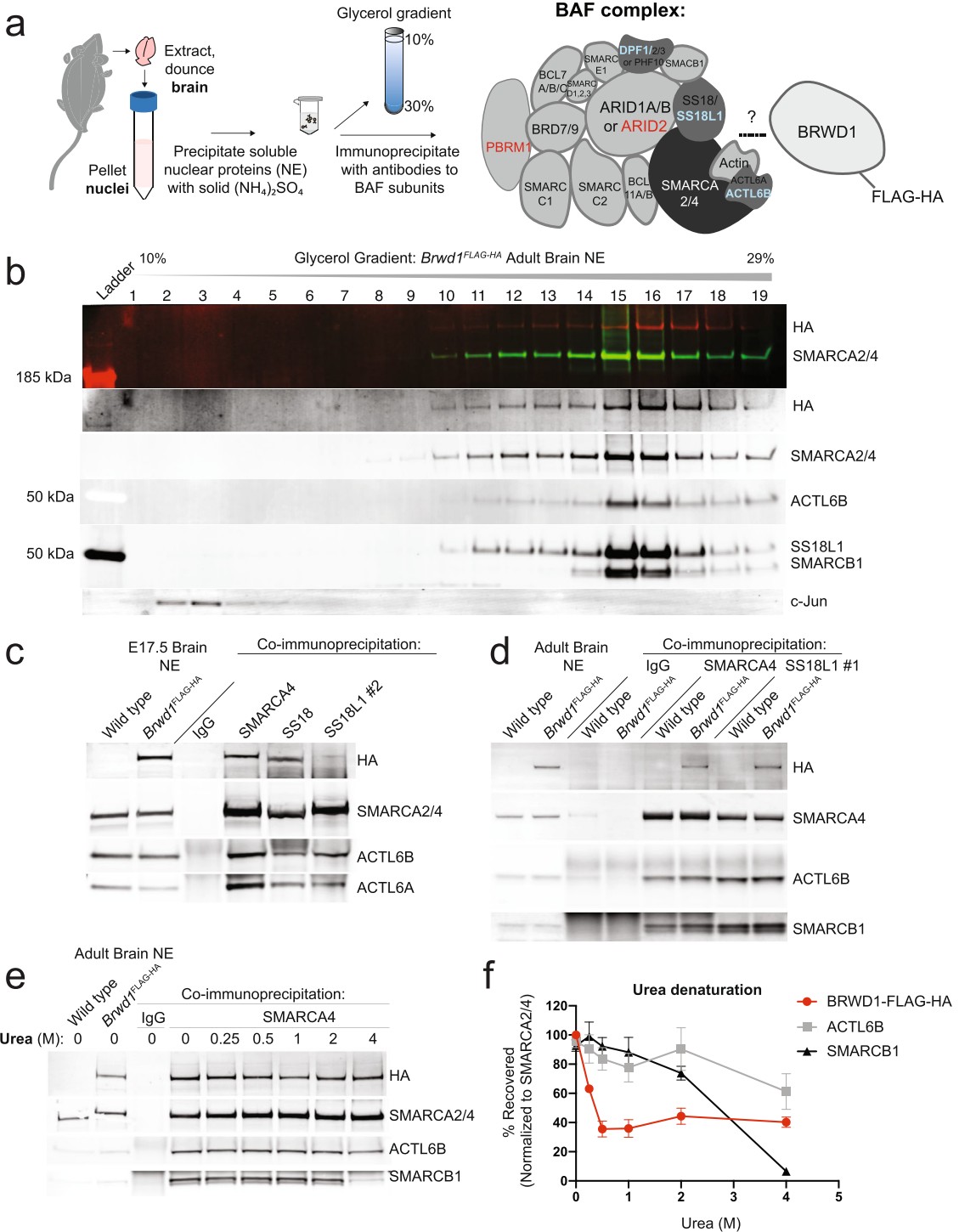

**Fig. 3 | BRWD1 tightly associates with the BAF complex in euploid brain.**
**a** Schematic of mouse brain soluble nuclear protein extract (NE) preparation, density sedimentation of nuclear proteins over a 10–30% glycerol gradient, and immunoprecipitation of BAF chromatin remodeling complexes. Blue lettering indicates neuronal-specific BAF subunits. Red lettering indicates PBAF-specific subunits. **b** Density sedimentation of adult *Brwd1*[FLAG-HA] brain NE over a 10–30% glycerol gradient indicates that BRWD1 predominantly associates with large protein complexes. Subunits of BAF and AP-1 complexes serve as molecular weight markers: SMARCA2/4 antibody indicates all BAF complexes including non-canonical GBAF (~1 MDa)[75], canonical BAF (~2 MDa) and Polybromo-containing BAF (PBAF, ~3 MDa); ACTL6B and SS18L1 indicate neuronal-specific BAF complexes; c-Jun indicates AP-1 (160–440 kDa). HA signal at the expected molecular weight of BRWD1-FLAG-HA (~260 kDa) is observed in fractions containing the BAF complex. **c** Endogenous BRWD1-FLAG-HA interacts with BAF complexes in embryonic brain.

BAF complexes were immunoprecipitated from *Brwd1*[FLAG-HA] brain NE with antibodies against the BAF core ATPase SMARCA4, the neural progenitor subunit SS18, the neuronal subunit SS18L1 or IgG as a control. Endogenous BRWD1-FLAG-HA robustly co-immunoprecipitated with SMARCA4 and the neural progenitor subunit SS18, but less so with the neuronal subunit SS18L1 from E17.5 brain. **d** BAF complexes purified from adult *Brwd1*[FLAG-HA] brain NE with antibodies against SMARCA4 or the neuronal subunit SS18L1 co-immunoprecipitate BRWD1-FLAG-HA. **e** The stability of the BAF:BRWD1-FLAG-HA interaction was challenged with increasing concentrations (0.25-4 M) of the denaturing agent, urea. A fraction of BRWD1 remained bound to BAF in up to 4 M urea, surpassing the stability of the dedicated BAF subunit, SMARCB1. **f** Quantification of urea denaturation experiments, as shown in **e**, with the amount of bound protein normalized to the amount of immunoprecipitated SMARCA4 ($n = 3$ experiments). Source data are provided as a source data file. See Supplementary Fig. 15 for uncropped blots with MW markers.

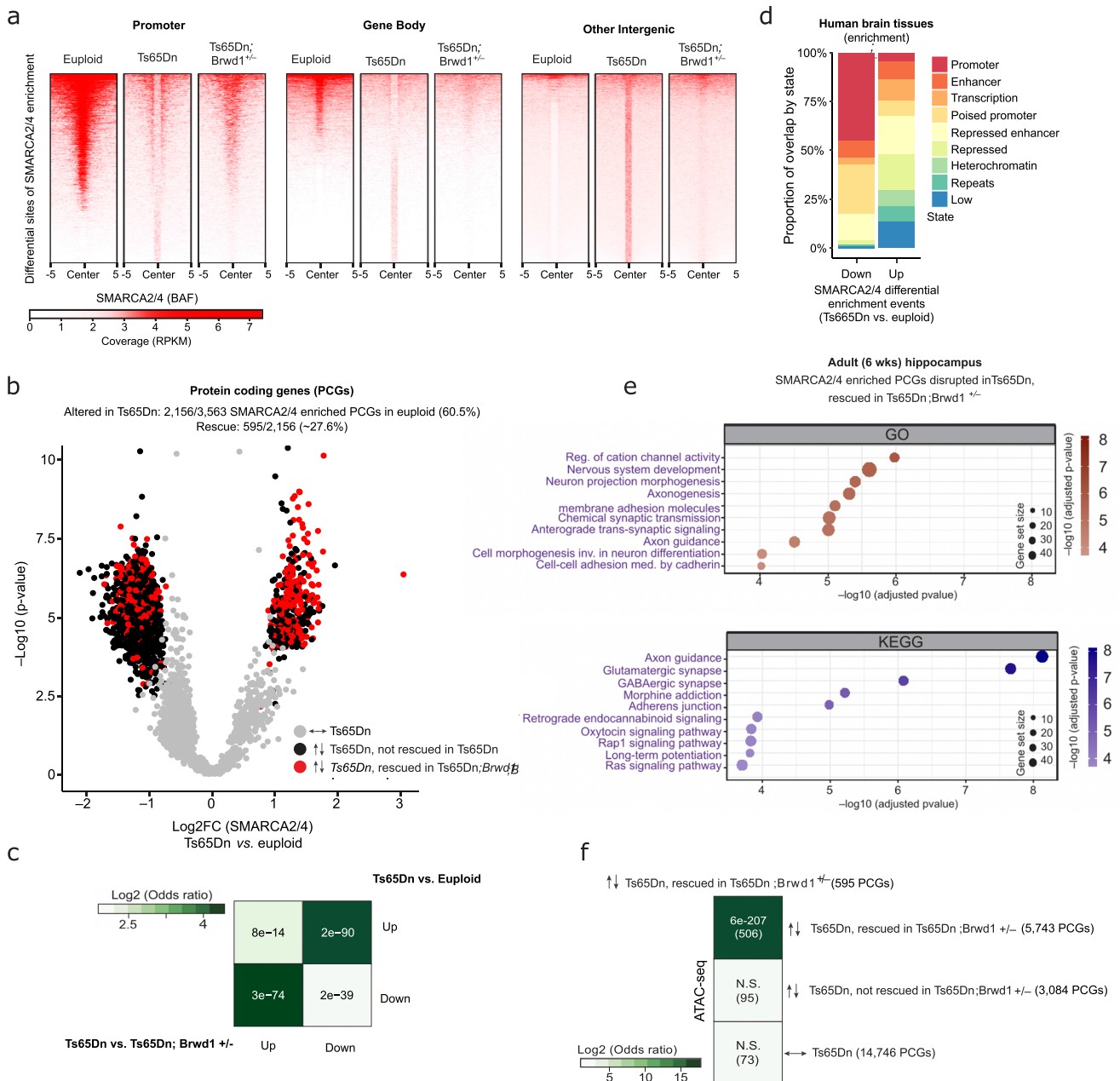

**Fig. 4 | Brwd1 renormalization partially rescues genomic BAF complex mistargeting in male trisomic brain. a** Heatmaps of normalized SMARCA2/4 enrichment in euploid vs. Ts65Dn vs. Ts65Dn;*Brwd1*[+/−] adult male (6-week) hippocampus centered (±5 kb) over sites of differential SMARCA2/4 enrichment comparing Ts65Dn vs. euploid mice, separated by genomic context. **b** Volcano plot depicting regulation of SMARCA2/4 enriched PCGs in euploid animals displaying differential enrichment in Ts65Dn mice; gray circles = unregulated PCGs. Of the PCGs regulated with respect to Smarca2/4 enrichment, 595 are rescued (red circles) in Ts65Dn;*Brwd1*[+/−] mice, whereas the remainder of PCGs do not display such rescue (black circles). **c** Relative frequency (observed/expected overlap in base pairs) of each chromatin state within significant differentially enriched sites for SMARCA2/4 (Ts65Dn vs. euploid). Chromatin states were obtained from brain regions included

in the Roadmap Epigenomics Project. **d** Bubble plots of GO terms (burgundy) and KEGG pathways (purple) displaying enrichment for rescued differentially enriched PCGs identified in **b** above. **e** Odds ratio analysis of overlapping differentially accessible sites in Ts65Dn vs. euploid animals and Ts65Dn;*Brwd1*[+/−] animals, separated by direction of regulation. **f** Odds ratio analysis of overlapping PCGs displaying rescued differential SMARCA2/4 enrichment in Ts65Dn vs. euploid animals vs. PCGs displaying differential neuronal chromatin accessibility in Ts65Dn vs. euploid mice that are either rescued, or not, in their differential accessibility in Ts65Dn;*Brwd1*[+/−] animals. Insert numbers indicate respective *p* values for associations, followed by the number of PCGs overlapping per category. See Supplementary Information Materials for full caption with *n*'s and statistics. Source data are provided as a source data file.

accessibility, indicating that BRWD1-mediated BAF mistargeting is associated with dysregulated chromatin structure (Fig. 4f).

## Discussion

In this study, we characterized roles for the epigenetic regulator, HSA21-encoded BRWD1, in DS-related phenotypes. Our findings demonstrated that elevated *Brwd1* expression is both necessary and

sufficient to precipitate DS-related impairments in cognition, synaptic physiology, and gene expression in Ts65Dn mice – deficits which were rescued by genetic renormalization of *Brwd1* copy number. We showed that BRWD1 stably interacts with the BAF complex in embryonic and adult brain and contributes to BAF mistargeting in adult Ts65Dn hippocampus. Approximately ~60% of BAF peaks were altered in trisomy, with many of these BAF binding events being

retargeted from promoters and poised genic regions towards inter-genic regions of the genome. BAF mistargeting was associated with changes in transcription and chromatin accessibility, implicating alterations in chromatin remodeling as a causative factor underlying DS-related impairments. Restoring *Brwd1* copy number rescued ~27% of BAF mistargeting in Ts65Dn hippocampus, indicating that BRWD1 is an important determinant of BAF genomic localization in brain.

We hypothesize that BRWD1 may act as a substoichiometric component for a subset of BAF complexes in brain, in that it functions as a histone-targeting protein for a certain fraction of total BAF (including both npBAF and nBAF) in a context-specific manner. Its substoichiometric relationship to BAF is supported by its dosage-dependent effects on BAF targeting, the fact that *Brwd1* deletion does not affect BAF assembly or mass on a glycerol gradient, and BRWD1's lower protein expression in mouse brain relative to other BAF subunits[36]. The observation that BAF was targeted away from "active" histone marks in trisomy does not necessarily mean that repressed regions are the primary target of BRWD1 in euploid neurons. In fact, BRWD1's bromodomain 2 was found to bind to transcriptionally-activating H3K14 and K18 acetylation in vitro[37], and its WD-repeat domain shares >84% identity with that of WDR5, which "reads" H3K4 as part of the MLL complex[38–40]. Whether BRWD1 directly binds histone modifications in brain, which histone marks it may bind to, and if its putative histone binding activity may contribute to BAF mistargeting in trisomy remain to be determined.

While no mouse model can fully recapitulate the complexity of the human DS disease state, the Ts65Dn line is unique amongst other commonly used DS models in that it carries triplicated MMU16 genes on a separate, freely segregating chromosome, *vs.* other models such as DP(16)1/Yey or Ts1Cje that harbor the HSA21-orthologous triplication as an extension or replacement on existing chromosomes[41]. Critically, this extra chromosome phenocopies the genomic instability caused by autosomal aneuploidy found in human DS, in addition to the HSA21 gene triplication. A recent study comparing three DS mouse models found that Ts65Dn exhibited features that most closely resembled the human symptomatic arc—particularly during embryonic time points, when Ts65Dn shows alterations in neuroanatomical features, cytoarchitecture (e.g., neocortical expansion/neurogenesis) and aberrant gene expression—while neither DP(16)1/Yey or Ts1Cje exhibited major prenatal symptoms[41]. Furthermore, in adults, Ts65Dn mice were found to display the most altered behavioral responses in hippocampal-dependent tasks and gene expression[41]. Interestingly, Ts65Dn males exhibited more profound developmental deficits and behavioral changes in adults compared with females, which paralleled aspects of sex differences observed in individuals with DS[41–44], as well as the sex differences observed in the current study. One notable drawback of the Ts65Dn model is that the additional MMU16 chromosome also contains the centromeric region of MMU17, including ~30 non-HSA21 orthologous PCGs. This complication of the Ts65Dn model highlights the importance of our experimental design utilizing a specific gene rescue approach that maintains the Ts65Dn trisomic genetic background in order to more precisely determine if non-HSA21 genes are driving the deficits observed in these mice.

Interestingly, in female Ts65Dn mice, we found that Brwd1 levels are more modestly increased *vs.* euploid controls and exhibit greater variability in displaying Brwd1 increases in comparison to males. Notably, BRWD1 has been previously shown to display sex-specific functions and expression levels, particularly in the context of early reproductive cell genesis[11]. Past studies have demonstrated that, in females, the effects of altered levels of HSA21 (e.g., *Dyrk1a*) may be mitigated by buffering mechanisms involving ncRNAs and other compensatory pathways[45]. Furthermore, while contextual fear learning depends, in part, on neural plasticity in the hippocampus, we were surprised to find that hippocampal LTP was only impaired in male Ts65Dn mice—an effect that was rescued by *Brwd1* normalization.

Consistent with this finding, our RNA-seq analyses directly comparing female *vs.* male Ts65Dn mice found that the most significant DE genes between the sexes in trisomy were associated with LTP. Furthermore, previous studies have found that male Ts65Dn mice have more severe hippocampal deficits than female mice, and that environmental enrichment improves female but not male Ts65Dn performance in spatial memory tasks[24]. In adult humans with DS, neurological symptoms are similar between males and females; however, male children with DS have been reported to display more externalizing behavioral deficits, including attention problems, thought problems, delinquent behavior and aggressive behavior[46]. Our results suggest that the mechanisms underlying cognitive impairments in Ts65Dn mice—and perhaps in DS individuals—may be different for males and females.

Additionally, it is important to consider that BRWD1 is not the only epigenetic regulator encoded on HSA21. Elevated expression of another HSA21-encoded gene, *DYRK1A*, has been observed in rodent models of DS and has been implicated in DS phenotypes[4,47]. Like BRWD1, DYRK1A overexpression can impair cognition in mice[48], and restoring *Dyrk1a* to euploid copy number (or pharmacologically inhibiting it) can rescue DS-related cognitive impairments[49,50], skeletal abnormalities[51,52] and Alzheimer's disease related phenotypes in trisomic mice[53]. Since BRWD1 and DYRK1A overexpression have strikingly similar effects on cognition, we hypothesize that they may function in the same pathway. DYRK1A also been shown to participate in BAF activity, most likely via its kinase activity. In support of this possibility, a study of phosphoproteins that showed treatment with a DYRK1A kinase inhibitor in Ts65Dn brain rescued altered phosphorylation of BAF complex subunits[48]. Importantly, our data suggest that BRWD1 may also contribute to BAF function through its binding to histone modification marks, thereby guiding the genomic targeting of BAF to facilitate chromatin restructuring. Therefore, our findings support a model in which BRWD1 renormalization may rescue the effects of BAF dysregulation, despite high levels of DYRK1A. How *BRWD1* may genetically interact with *DYRK1A* or other HSA21-encoded proteins to regulate cognition is an important avenue for future research. In future studies aimed at validating and investigating DS-related mechanisms in models for human brain development, such as cerebral organoids, it will be important to evaluate the effects of single allele mutations for key HSA21 genes (e.g., *BRWD1*) in DS patient-derived systems and examine DS-related molecular signatures for evidence of genetic contributions and interactions in DS.

In conclusion, we demonstrated that a previously uncharacterized binder of the BAF complex, BRWD1, is critically involved in regulating aberrant neuronal gene expression patterns in DS-like brain, which precipitate synaptic and cognitive deficits associated with this disorder. Increased *Brwd1* expression results in BAF genomic mistargeting, inappropriate patterns of chromatin accessibility and dysregulated gene expression contributing to aberrant plasticity. Gaining a better understanding of BRWD1's precise functions within the BAF complex, as well as its potential histone "reader" activities, will greatly improve our knowledge of the molecular underpinnings of DS.

## Methods

### Cell culture

**hiPSCs, NPCs, and neurons.** Reprogrammed, de-identified fibroblasts from one mosaic Trisomy 21 (Ts21) patient (line AG05397)—yielding Ts21 hiPSCs and an isogenic control—were provided by Dr. Anita Bhattacharyya (UW–Madison) and were generated and karyotyped, as previously described[6]. Rosettes were cultured in NPC medium (DMEM/F12, 1x N2, 1x B27-RA (Life Technologies), 1 µg ml⁻¹ Laminin and 20 ng ml⁻¹ FGF2) and dissociated in TrypLE (Life Technologies) for 3 min at 37 °C. NPCs were maintained at high density, grown on poly-ornithine/laminin or Matrigel (BD) coated plates in NPC medium and split approximately 1:4 every week with Accutase (Life Technologies)[54,55]. For neuronal differentiations, NPCs were

dissociated with Accutase and plated in neural differentiation medium (DMEM/F12, 1× N2, 1× B27-RA, 20 ng ml$^{-1}$ BDNF (Peprotech), 20 ng ml$^{-1}$ GDNF (Peprotech), 1 mM dibutyryl-cyclic AMP (Sigma), 200 nM ascorbic acid (Sigma) onto poly-ornithine /laminin-coated plates and matured for 4 weeks.

**ANIMALS.** All mice used in DS related studies exist on a mixed genetic background, with comparative groups (euploid *vs. Brwd1$^{+/-}$ vs.* Ts65Dn vs. Ts65Dn;*Brwd1$^{+/-}$*) maintained on the same mixed background (DBA/2J x B6EiC3Sn/J) for experimental testing. Briefly, trisomic *Ts(17⁶)65Dn* (Ts65Dn) females (Jackson Labs Stock 001924) were crossed to B6EiC3Sn/J (Stock 001875) euploid males to generate euploid and Ts65Dn animal littermates for initial *Brwd1* expression analyses. *Brwd1$^{repro5}$* mutant mice (Mouse Genome Informatics [MGI] ID 3512929)[11] were kindly provided by Dr. John Schimenti (Cornell). Heterozygotic *Brwd1$^{repro5}$* mutant male mice, aka *Brwd1$^{+/-}$* (fully backcrossed > 10 generations to B6EiC3Sn/J to match the breeding scheme for euploid vs. Ts65Dn animals), were crossed to Ts65Dn females to generate euploid *vs. Brwd1$^{+/-}$* vs. Ts65Dn *vs.* Ts65Dn;*Brwd1$^{+/-}$* animals for genetic rescue experiments. All animal protocols were approved by the IACUC at the Icahn School of Medicine at Mount Sinai (ISMMS).

**Generation of Brwd1$^{FLAG-HA}$ mice by CRISPR/Cas9 genome editing**
Given that suitable antibodies for the detection of endogenous Brwd1 in rodent tissues are not commercially available, we generated *Brwd1$^{FLAG-HA}$* mice for in vivo interrogations. An optimal guide sequence was selected using online software at mit.crispr.edu. The selected guide RNA sequence corresponding to *Brwd1* was as follows: CAGCCTACTCCGAGG. The DNA template for making sgRNA was generated using a cloning-free overlap PCR method, essentially as described[56]. The DNA template was reverse transcribed into RNA using an Ambion MEGAshortscript T7 Transcription Kit (cat#AM1354), then purified using Qiagen MinElute columns (cat#28004). For pronuclear injection, the sgRNA (50 ng/μL), ssODN (50 ng/μL, IDT Ultramer Service), and Cas9 mRNA (25 ng/μL, TriLink) were co-injected into zygotes (F1 hybrids between strains FVB/NJ and B6(Cg)-Tyrc-2J/J) then transferred into the oviducts of pseudopregnant females. Founders carrying at least one copy of the desired alteration were identified and backcrossed into FVB/NJ. Initial phenotyping was done after one backcross generation, and additional phenotyping was done with mice backcrossed at least two or more generations (Supplementary Fig. 10).

| | PURPOSE | SEQUENCE (5′ to 3′) |
|---|---|---|
| ssODN with 1xFLAG and 1xHA | CRISPR/Cas9 | ATCTTAGGCGGTTCAG ATCCCGGAAGGAAAAAGCCC AGCCTACTCCGAGG **GACTACAAA GACGATGACGACA AG**GGG**TATCCCTATG ACGTCCCGGACTATGCA** TAGAAAGGTTACCGGG AATTGTCAGCAGCTCC AATGCCTGCCCTGAAGTC |
| sgRNA Template (T7 Forward) | CRISPR/Cas9 | GAAATTAATACGACTCACTAT AGGCAGCCTACTCCGAGGTA GAAGTTTTAGAGCTA GAAATAGC |
| sgRNA Template (T7 Reverse) | CRISPR/Cas9 | CAAAATCTCGATCTTT ATCGTTCAATTTTATTCCGAT CAGGCAATAGTTGAACT TTTTC ACCGTGGC TCAGCCACGAAAA |
| Brwd1 Common Primer (Forward) | Genotype, RT-qPCR | TGCATAGTGACACCCTGAAAG |
| Brwd1 Common Primer (Reverse) | Genotype, RT-qPCR | 3′-GCTGTGTA-GAGCTAACTGGAAG-5′ |

| | | |
|---|---|---|
| Epitope-Specific Primer (Reverse) | RT-qPCR | 3′-TCCGGGACGTCA-TAGGGATA-5′ |
| Brwd1 repro5 Primer (Forward) | Genotype | ATGGCCACTGTAGGTTCAGC |
| Brwd1 repro5 Primer (Reverse) | Genotype | TTAAAGTCCACGACCCCTGA |

**Sequences of primers and oligonucleotides**
**Brwd1$^{FLAG-HA}$ genotyping.** Crude lysates for PCR were made from small tissue biopsies (tail, toe or ear punches). Genotyping primers are listed above. PCR reactions were carried out as follows: initial denaturation at 95° for 5 min, then 30 cycles of 95° for 30 s, 58° for 30 s, 72° for 30 s, and a final elongation at 72° for 5 min. For identification of *Brwd1$^{FLAG-HA}$*, amplicons were analyzed on high percentage agarose gels. The WT allele yields a band at 454 bp, and the FLAG-HA allele yields a band at 508 bp.

### Housing and oversight
Mice were group housed (separated by sex)—with the exception of surgerized animals, which were singly housed post-surgery—under a 12-h-light/dark cycle at constant temperature (25 °C) and humidity with ad libitum access to food and water. Animals arriving from external sources were allowed at least one week of habituation to housing conditions prior to experimentation. Both male and female mice were assessed in these studies [embryonic/E17.5 (mixed male and female) and adult (6-week, males and females analyzed separately). All procedures were performed in accordance with NIH guidelines and the Institutional Animal Use and Care Committees (IACUC) at the Icahn School of Medicine at Mount Sinai, Michigan State University and Cornell University.

**RNA isolation and qPCR**
hiPSC neurons were collected following differentiation (4 weeks) and pellets were immediately homogenized in RLT buffer (Qiagen). Forebrain from E17.5 embryos, and whole mPFC, hippocampus and cerebellum were collected from 6-week-old mice and flash frozen. For validation of *Brwd1* overexpression in viral transduction experiments, mice were euthanized 4 days following viral infusion and brains were flash frozen. Infected brains were then sectioned at 100 μm on a cryostat and GFP was illuminated using a NIGHTSEA BlueStar flashlight to microdissect virally transduced tissues. All tissues were homogenized in Trizol (Thermo Fisher). RNA was isolated on RNeasy Microcolumns (Qiagen) following the manufacturer's instructions. RNA was then reverse transcribed using iScript (BioRad cat# 1708891) and cDNA was quantified by qPCR using SYBR Green (Qiagen). Each sample was run with technical replicates and gene expression fold changes were calculated using the ΔΔCt method normalized against the housekeeping gene *GAPDH/Gapdh*. For *Brwd1* rescue validations, Brwd1_3′ primers were used, and for all other mouse assessments, Brwd1 primers were used. Sequences of qPCR primers (forward and reverse) used are as follows:

Mouse
GapdhF: AGGTCGGTGTGAACGGATTTG
GapdhR: TGTAGACCATGTAGTTGAGGTCA
Brwd1_3′F: GCCTGGTGTTCAGATGCTGTG
Brwd1_3′R: GCTGTTCCATCTCGGCTACCA
Brwd1F: TGAGTGATGCAGAGGATTCG
Brwd1R: TGCTGTTGTGGACAGAATGG
FosF: GAACGGAATAAGATGGCTGC
FosR: TTGATCTGTCTCCGCTTGG
Npas4F: CTGCATCTACACTCGCAAGG
Npas4R: GCCACAATGTCTTCAAGCTCT
Zif268F: ACCACAGAGTCCTTTTCTGAC
Zif268R: AAGCGGCCAGTATAGGTGATG
BdnfF: TCAGCAGTCAAGTGCCTTTG
BdnfR: TCAGTTGGCCTTTGGATACC

Human
GAPDHF: AATCCCATCACCATCTTCCA
GAPDHR: TGGACTCCACGACGTACTCA
BRWD1F: AGCCCTTTGCACTCGTTATG
BRWD1R: GGGTTTCAGTTGGCACAATC

For initial validations of the *Brwd1*[FLAG-HA] line, testes were harvested from 8-week-old wildtype, heterozygous and homozygous tagged mice, and tissues were homogenized. In brief, total RNA was extracted and purified using Trizol following the manufacturer's instructions. Next, 1.5ug of RNA from each condition was reverse transcribed into cDNA using qScript cDNA SuperMix kit (Quantabio; cat# 101414-102). The resulting cDNA was used as a template and combined with Fast SYBR Green Master Mix (Life Technologies) for qPCR. Custom primers were designed using Primer3 and are listed in the Table above. Assays were run on a CFX96 Touch™ Real-Time PCR Detection System (BIO-RAD), where each sample was run in triplicate. The Ct values were obtained and averaged per triplicate reaction and then normalized to *GAPDH*.

## Immunocytochemistry

Human cells were fixed in 4% paraformaldehyde in PBS at 4 °C for 10 min, permeabilized at room temperature for 15 min in 0.1% Triton in PBS and then blocked in 5% donkey serum with 0.1% Triton at room temperature for 30 min. The following primary antibodies and dilutions were used: goat anti-NANOG (R&D), 1:200; mouse anti-TRA1-60 (Chemicon), 1:100; mouse anti-human NESTIN (Chemicon), goat anti-SOX2 (Santa Cruz), 1:200; rabbit anti-βIII-tubulin/Tuj 1 (Covance), 1:200; mouse anti-MAP2AB (Sigma), 1:200. Secondary antibodies used include Alexa donkey 488 and 568 anti-rabbit (Life Technologies), Alexa donkey 488 and 568 anti-mouse (Life Technologies), and Alexa donkey 488 and 568 anti-goat (Life Technologies); all were used at 1:300. To visualize nuclei, slides were stained with 0.5 µg ml−1 DAPI (4′,6-diamidino-2-phenylindole) and then mounted with Vectashield.

## Field electrophysiology

Acute slices were prepared from 6-week-old mice, as previously described[57,58]. Briefly, mice were deeply anesthetized with isoflurane, and their brains were rapidly removed and immersed in an ice-cold modified ACSF solution containing: 215 mM sucrose, 2.5 mM KCl, 1.6 mM $NaH_2PO_4$, 4 mM $MgSO_4$, 1 mM $CaCl_2$, 4 mM $MgCl_2$, 20 mM glucose, and 26 mM $NaHCO_3$ (pH = 7.4, equilibrated with 95% $O_2$/5% $CO_2$). Coronal brain slices (400-µm thick) containing the hippocampus were prepared with a vibrating slicer (VT1000S; Leica Microsystems) and then incubated at room temperature for ≥3 h in physiological ACSF, containing (in mM): NaCl (120), KCl (3.3), $Na_2HPO_4$ (1.2), $NaHCO_3$ (26), $MgSO_4$, (1.3) $CaCl_2$ (1.8) and glucose (11), equilibrated to pH 7.4 with 95% $O_2$ and 5% $CO_2$. Hemi-slices were transferred to a submersion recording perfused with ACSF at a flow rate of ~2 mL/min using a peristaltic pump; experiments were performed at 28.0 ± 0.1 °C. Recordings were acquired with a GeneClamp 500B amplifier and Digidata 1440A digitizer (Molecular Devices), with all signals low-pass filtered at 2 kHz and digitized at 10 kHz. Field excitatory postsynaptic potentials (fEPSPs) were recorded with a patch-type pipette filled with ACSF ($R_e$ = 2–3 MΩ), positioned in the middle third of stratum radiatum in area CA1. fEPSPs were evoked by 60-µs square-wave monophasic stimuli generated by an ISO-Flex stimulus isolator (A.M.P.I.) and delivered to the Schaffer collaterals by a concentric bipolar electrode (FHC) positioned in the middle third of stratum radiatum, 150–200 µm away from the recording pipette. Input-output curves were generated by a series of stimuli in 0.1-mA steps. Paired-pulse ratios (PPR) were determined by delivering two stimuli at intervals of 20, 50, and 100 ms; each interstimulus interval was repeated three times, and the resulting potentials were averaged. PPR was calculated as slope of the second fEPSP divided by slope of the first fEPSP. Long-term potentiation (LTP) was induced after 20 min of stable baseline recordings (at 0.033 Hz) by theta-burst stimulation (TBS), which consisted in a series of 10 bursts of 4 stimuli (100 Hz within the burst, 200-ms interburst interval), repeated four times (10 s apart), and delivered at a stimulus intensity that produced a baseline response of 75% of spike threshold. Following TPS, stimulation at 0.033 Hz resumed for 60 min. All results were analyzed by ANOVAs followed, where appropriate, by Tukey post-hoc tests.

## Patch electrophysiology

All recordings were carried out blind to the experimental conditions. Male C57BL/6 3-month-old mice were injected intra-CA1 with either HSV-GFP or HSV-BRWD1-GFP. After 4–5 days of expressing HSV-GFP vs. HSV-BRWD1-GFP, mice were perfused with cold artificial cerebrospinal fluid (aCSF) containing (in mM): 128 NaCl, 3 KCl, 1.25 NaH2PO4, 10 D-glucose, 24 $NaHCO_3$, 2 $CaCl_2$, and 2 $MgCl_2$ (oxygenated with 95% $O_2$ and 5% $CO_2$, pH 7.35, 295–305 mOsm). Acute brain slices containing CA1 were cut using a microslicer (DTK-1000, Ted Pella) in sucrose-ACSF, which was derived by fully replacing NaCl with 254 mM sucrose, and saturated by 95% $O_2$ and 5% $CO_2$. Slices were maintained in the holding chamber for 1 h at 37 °C. Slices were transferred into a recording chamber fitted with a constant flow rate of aCSF equilibrated with 95% $O_2$/5% $CO_2$ (2.5 ml/min) and at 35 °C. Glass microelectrodes (2–4 MΩ) filled with an internal solution containing (mM): 115 potassium gluconate, 20 KCl, 1.5 $MgCl_2$, 10 phosphocreatine, 10 HEPES, 2 magnesium ATP and 0.5 GTP (pH 7.2, 285 mOsm). All recordings were performed in GFP labeled cells located in the CA1 pyramidal layer. Cell excitability of CA1 neurons expressing GFP was measured with 2 s incremental steps of current injections (50, 100, 150, and 200 pA) at −70 mV holding potential. Series resistance was monitored during all recordings at the beginning and end of each recording, and data were rejected if values changed by more than 20%. All data acquisition and on-line analysis were collected using 700B amplifier, Digidata 1322 A digitizer, and pClamp 10.2 (Molecular Devices). Spontaneous excitatory postsynaptic currents (sEPSCs) were recorded in voltage clamp at a holding potential of −70 mV with series resistance of <6 MΩ, in the presence of picrotoxin (50 µM). For recording sEPSCs, the external aCSF solutions contained 50 µM dl-2-amino-5-phosphonovaleric acid (AP-5) to block NMDA receptors. The ionic composition of the internal (pipette) solution for voltage-clamp studies of sEPSC consisted of (in mM) 140 CsCl, 10 phosphocreatine, 2 $MgCl_2$, 10 EGTA, 2 magnesium ATP and 0.5 GTP and 10 HEPES with a pH adjusted with CSOH. sEPSC was analyzed with the MiniAnalysis software (Synaptosoft). Briefly, sEPSCs were detected automatically using an amplitude threshold of 10 pA and then visually accepted or rejected based upon the rise and decay times. Spontaneous inhibitory postsynaptic currents (sIPSCs) were recorded in aCSF containing kynurenic acid (3 mM), with the holding potential set at 0 mV. To compute the average sIPSCs, the synaptic events with single-peaks were aligned with the rise time. The amplitude of GABA-activated tonic current was measured as the difference in the baseline holding current before and after the application of bicuculline.

## Behavior

**Fear conditioning (FC).** Mice (6–8 weeks old) were habituated to the testing room for 20 min prior to training. Mice were then trained over three conditioning trials, each consisting of random intervals with a 2.0-s, 0.6-mA foot shock. Training in context A was conducted in a dark room with white noise. Context A chambers consisted of a square plexiglass chamber with metal rods (which delivered the shock during training), and the floor had been washed with 70% ethanol. Context B (no training) was conducted in white light with no noise and consisted of a cylinder plexiglass chamber with a flat white plastic floor that had been washed with Micro-90. Context B chambers had a solution of 10% vanilla extract placed next to them.

Testing for conditioned fear responses (freezing) to the trained context occurred 24 h following training. Mice were equally divided and placed into either Context A or Context B for 5 min and total seconds of freezing were recorded (EthoVision). Mice were then exposed to the opposite context 4 h following the first testing session and freezing was recorded. Freezing was expressed as a percentage of the total test time.

**Temporally dissociated passive avoidance (TDPA).** Passive avoidance learning was assessed in mice, modified from a published protocol[59]. For conditioning, mice were placed into the lighted side of a divided light-dark chamber (Coulbourn Instruments). After 1 min, the entry door to the dark side was raised and latency to crossover to the dark side was recorded. Entry was defined as whole body, including all four paws and tail base, on the dark side. Upon an entry, the door was closed. 30 s later, mice were administered a mild electric footshock (0.8 mA, 2 s duration), and following another 30 s, mice were returned to their home cage. Mice were tested the following day. Testing was conducted in the same manner as conditioning. When a criterion of 300 s spent in the light side was reached, a mouse would be returned to its home cage.

**Open field.** For locomotion and anxiety-like behaviors, mice were placed in an open field apparatus (Omnitech Electronics, Inc., Columbus, OH), which consisted of a $16 \times 16$-in. plastic chamber surrounded by 16 photobeam detectors along the $x$- and $y$-axis to measure horizontal movement. Mice were placed in the open field apparatus for 30 min and distance and time in center vs. periphery were recorded. Behavior was analyzed using Fusion software (Version 5.6, Omnitech Electronics, Inc.).

**Elevated plus maze (EPM).** Anxiety-like behavior was assessed in mice conducted using the EPM test[60]. Mice acclimated to the maze room for 30 min prior to testing. During testing, mice were placed into the center of the maze and allowed to explore for 5 min before being placed back into their home cage. Behavior was measured using automated videotracking software (CleverSys, Inc.).

**Sucrose preference.** A 2-bottle choice test protocol was used for assessing sucrose preference[61]. Mice were single-housed for 24 h prior to testing and then throughout the experiment. Custom bottles (2 per cage) containing RO water were first placed into the cages for 4 days for a baseline assessment of bottle preference. For the testing period, one of the bottles was replaced with a 2% sucrose solution in RO water for 4 more days. Bottle weights were measured daily every morning (9 A.M.) throughout the experiment, switched sides (left or right side of cage top) daily, and preference for the sucrose-containing bottle was assessed.

**Morris water maze (MWM).** Water maze testing was conducted as previously described[60]. Briefly, swim behavior was measured in a water pool filled with white beads and a platform. Prior to surgeries, to acclimate mice to the water maze, animals were trained to find a visible, cued platform randomly placed at different locations for six 60 s trials. After surgeries, training was conducted across 4 daily trials (1 h intertrial intervals) for 5 days to locate a hidden platform. Trials lasted 60 s or when the mouse reached the platform, whichever came first. 24 h after the last day of training, a 60 s probe test was conducted, where the platform was removed. Trial latency and swim speed (cm/s) were recorded during both training and testing. Quadrant time was recorded during the probe test.

### Generation of HSV viral constructs
The mouse BRWD1-FLAG-HA coding sequence was subcloned into the bicistronic p1005+ HSV plasmid expressing GFP under the control of the human immediate early cytomegalovirus promoter (CMV). An IE4/5 promoter drives BRWD1-FLAG-HA expression. HSV-GFP vectors were used as controls in behavioral experiments. Viral constructs were packaged at the Gene Technology Core (Massachusetts General Hospital).

We used HSV vectors in our studies of sufficiency for the following reasons: 1) HSVs solely infect neuronal cell bodies within an injected region of brain; 2) HSV-encoded transgenes are expressed very rapidly (within 12 h) but only transiently (they dissipate within 7 days); and 3) it was necessary to use HSV vectors to express BRWD1, since the *Brwd1* gene far exceeds the maximum insertion size for other neuronal specific vectors, such as AAVs. While complementary to our copy number restoration studies, the expression of *Brwd1* per HSV-infected neuron likely exceeds that of the trisomic context, a phenomenon that should be considered when comparing phenotypes arising from the two manipulation strategies.

**Novel environment (NE).** For immediate early gene (IEG) qPCR assessments in Fig. S4A, microdissected tissues from mice transduced with HSV-GFP vs. HSV-BRWD1-GFP (CA1) were collected from animals either in their home cage (HC) or after being moved to a NE for 30 min vs. 90 min.

### Stereotaxic surgery and viral delivery
Animals were anaesthetized with a ketamine/xylazine solution (100/10 mg/kg) i.p. and then positioned in a stereotaxic frame (Kopf instruments). HSV-BRWD1 or HSV-GFP were bilaterally infused (0.5 μL at 0.1 μL/min; 7° angle) into dorsal hippocampus using the following coordinates: −2.2 AP, ±2.0 ML, −2.0/−1.8 mm from bregma. All tissue collections and/or behavioral experiments commenced 24 h to 4 days after surgery.

### Immunohistochemistry (IHC)
For viral *Brwd1* overexpression studies, mice were anesthetized with a ketamine/xylazine solution (100/10 mg/kg) 4 days following viral infusions and transcardiacally perfused with cold phosphate-buffered saline (PBS)/4% paraformaldehyde (PFA). Brains were post-fixed overnight (-12 h) in 4% PFA and then cryoprotected in 30% sucrose/PBS 1× for 2 days at 4 °C. Brains were sectioned at 40 μm on a cryostat. Brain sections were then incubated overnight at room temperature with primary antibodies. Antibodies used for hippocampal brain sections: anti-chicken GFP. The following day, brain sections were washed 3× in 1× PBS and then incubated for 2 h at room temperature with a fluorescent-tagged Alexa Fluor antibodies, washed 3× in 1× PBS and then incubated with DAPI (1:10,000) for 5 min at room temperature. Brain sections were then mounted onto slides with Prolong Gold and immunofluorescence was visualized using a confocal microscope (Zeiss LSM 780).

**Double labeling of HA-tag with DDX4 in mouse testis.** 4-μm–thick sections of paraformaldehyde-fixed/paraffin-embedded mouse testis were used for immunohistochemical analyses. After deparaffinization in xylene and rehydration in graded ethanol, heat-activated antigen retrieval was performed in Tris-EDTA (pH9.0) for 20 min. Endogenous peroxidase activity was quenched with 0.3% hydrogen peroxide in distilled water for 10 min. The HA-tag and DDX4 IHC detections were performed using an ImmPRESS HRP Anti-Rabbit Ig (Peroxidase) Polymer Detection Kit and an ImmPRESS™-AP Anti-Rabbit IgG (alkaline phosphatase) Polymer Detection Kit, respectively (Vector Laboratories), following the kit instructions. Rabbit anti-HA-tag antibody was used at 1:50 and labeled with DAB, while rabbit anti-DDX4 was used at 1:1000 and labeled with Red substrate. Finally, tissues were counterstained with hematoxylin and mounted with Permount. IHC results were examined by Olympus AX 70 compound microscope equipped with MicroFire camera and PictureFrame for image processing and capture (Optronics).

## Primary cortical neuronal culture

Pregnant female mice were euthanized and cortices from E16 embryos were dissected. Tails from each embryo were collected for genotyping. Cortices were incubated individually in trypsin-EDTA (0.25%; Gibco) for 10 min to dissociate and neurons were plated at 0.6 million cells per well in six-well Poly-D-Lysine (Sigma) coated plates in DMEM (Gibco) containing 10% fetal bovine serum and 1% penicillin–streptomycin. The following day, the medium was changed to a serum-free medium containing neurobasal and B27 (Gibco) and AraC to inhibit glial cell proliferation. Half media changes occurred every 4 days. Cells were maintained at 37 °C, 5% $CO_2$, and 95% humidity. Cells were collected at DIV12 for RNA sequencing.

## RNA-SEQ library preparation and sequencing

Following RNA purification with trizol and the Qiagen RNAeasy Mine-lute kit (Cat.# 74204), libraries were prepared using the TruSeq RNA Library Prep Kit v2 according to Illumina protocols, multiplexed library sizes were validated on an Agilent Bioanalyzer system and then sequenced (single-read) on an Illumina HiSeq 4000 or Illumina Nova-Seq 6000 sequencer.

## RNA-SEQ analyses

Raw sequencing reads from mouse embryonic forebrain, e17.5 primary neuronal cultures, or adult hippocampus were mapped to mm10 using HISAT2(Version 2.2.1 + galaxy0)[62]. Counts of reads mapping to genes were obtained using featureCounts(v2.0.1 + galaxy2)[63] against Ensembl v90 annotation. Read counts were normalized using RUVr (v1.24.0), and differential expression analysis was done using the DESeq2 package (v1.6.3)[64]−for both likelihood-ratio-test and pairwise comparisons −at FDR cutoffs of 0.1. GO analyses were conducted using iDEP[Ge, 2018 #12]. Odds ratio analyses between DE gene lists and human DS RNA-seq[20] were conducted based on gene names using geneOverlap (*GeneOverlap: Test and visualize gene overlaps*. R package version 1.23.0).

## Experiments related to BAF-BRWD1 complex associations

**Ammonium sulfate precipitation of soluble brain nuclear proteins.**
Adult or E17.5 mouse brains were extracted, flash-frozen in liquid nitrogen and stored at −80 °C. Brains were thawed in 10 mL of ice-cold Buffer A [10 mM HEPES, pH 7.5, 25 mM KCl, 1 mM EDTA, 0.1% NP40, 10% glycerol, plus 1 mM DTT, protease inhibitor tablet (Roche), 1 mM sodium orthovanadate, and 10 mM sodium butyrate freshly added at time of use]. Nuclei were released from cells by douncing brain tissues with ten strokes of a loose-fitting pestle, followed by ten strokes with a tight-fitting pestle on ice. The lysates were then centrifuged at 1700 × *g* for 10 min at 4 °C to pellet nuclei. The nuclei were washed 2× with 5 mL ice-cold Buffer A and then washed 1× in 3 mL Buffer C [10 mM HEPES pH 7.5, 100 mM KCl, 1 mM EDTA, 3 mM $MgCl_2$, 10% glycerol plus 1 mM DTT, protease inhibitor tablet (Roche), 1 mM sodium orthovanadate, and 10 mM sodium butyrate freshly added at time of use]. The supernatant was removed from the pelleted nuclei and the volume was adjusted to 2.8 mL exactly with Buffer C. The resuspended nuclei were then divided evenly into four 1.5 mL microcentrifuge tubes. To each 700 µL of resuspended nuclei, 77.7 µL of 3 M ammonium sulfate was added drop-wise to lyse nuclei and salt out the DNA. The solutions were rotated at 4 °C for 30 min to overnight (overnight worked slightly better). DNA was pelleted by ultracentrifugation (Beckman Coulter) at 100,000 rpm for 15 min at 4 °C. The supernatant containing soluble nuclear proteins was transferred to a clean 1.5 mL microcentrifuge tube and 233 mg of solid ammonium sulfate was added to precipitate soluble nuclear proteins. The mixture was rotated for 20 min to overnight at 4 °C. The precipitated nuclear proteins were pelleted by ultracentrifugation at 100,000 rpm for 20 min at 4 °C. Pelleted nuclear proteins from brain tend to float on top of the buffer and so the buffer

must be carefully removed. Nuclear extracts (4 tubes/brain) were then stored at −80 °C.

**Density sedimentation (10–30% glycerol gradient).** Brain nuclear extract from one quarter to half of an adult mouse brain (wild type, *Brwd1*[−/−] or *Brwd1*[*FLAG-HA*]) was resuspended in 220 µL ice cold HEMG-0 buffer (50 mM HEPES pH 7.9, 100 mM KCl, 0.1 mM EDTA, 12.5 mM $MgCl_2$, plus 1 mM DTT, protease inhibitor tablet (Roche), 1 mM sodium orthovanadate, and 10 mM sodium butyrate freshly added at time of use). Of this, 10% was reserved as input. In the meantime, a 10 mL, 10–30% gradient of glycerol in HEMG buffer was poured into a 14 × 89 mm polyallomer centrifuge tube (Beckman Coulter cat. # 331372). The resuspended nuclear extract was carefully laid atop the gradient and then centrifuged in an SW41 rotor at 40,000 rpm for 16 h at 4 °C. Twenty 0.5 mL fractions were carefully collected and 10% of each fraction was run on an SDS PAGE gel. Nuclear proteins were transferred overnight (26 h was optimal for 260 kDa BRWD1) at 85 mA (constant amperage) to a PVDF membrane and then immunoblotted using antibodies to subunits of the BAF complex or to HA (see Antibodies).

**Co-immunoprecipitations.** Antibodies raised against subunits of the BAF complex (see Antibodies) were bound to Protein G Dynabeads (ThermoFisher cat. # 10009D) at a ratio of 8 µg antibody: 50 µL Dynabeads per IP. The unbound antibody was removed and the beads were washed 3× in immunoprecipitation (IP) buffer [20 mM HEPES pH 7.5, 150 mM KCl, 1 µM $CaCl_2$, 1 mM $MgCl_2$, 0.1% Triton X-100, 10% glycerol, plus 1 mM DTT, protease inhibitor tablet (Roche), 1 mM sodium orthovanadate, and 10 mM sodium butyrate freshly added at time of use]. Nuclear protein pellets were resuspended in 220 µL IP buffer and the protein concentration was determined by Bradford assay. For each IP, 250–350 µg of nuclear extract was resuspended to 0.25 mg/mL and added to the antibody-bound beads (note that the dilute concentration of 0.25 mg/mL is critical to prevent non-specific binding of brain proteins to the beads). The nuclear extract was rotated overnight at 4 °C. The next day, the supernatant or "flow-through" was separated and reserved for depletion studies (see below). The beads were washed 5× with ice cold IP buffer. Nuclear proteins were eluted by boiling in 44 µL 1.1× LDS Sample Buffer (Invitrogen) supplemented with β-mercaptoethanol, then separated on an SDS-PAGE gel and transferred overnight (26 h ideal for BRWD1) at 85 mA (constant amperage) to a PVDF membrane. Antibodies raised against subunits of the BAF complex or against HA (to indicate endogenously tagged BRWD1) were used to detect immunoprecipitated proteins.

**Depletion studies.** To roughly determine how much BRWD1-FLAG-HA was associated with the BAF complex in brain, we successively immunoprecipitated BAF complexes with α-SMARCA4, α-SS18 or α-SS18L1 antibodies, or IgG as a control, three times from the same brain nuclear extract. Specifically, the flow-through from the initial IP experiment was used as input for the second IP, and the resulting flow-through was used as input for the third IP. The ratio for each immunodepletion was approximately 1 µg antibody for each 31 µg nuclear extract. A fraction of the final flow-through representing three immunodepletions was run on an SDS PAGE gel and transferred to a PVDF membrane as above. Subunits of the BAF complex, BRWD1-FLAG-HA, and control proteins (TBP and actin) were detected with specific antibodies (see Antibodies). The intensity of protein bands was quantified by densitometry using ImageJ. Signal remaining in the nuclear extract after immunodepletion was normalized to the average signal in the IgG ($n = 4$ biological replicates for adult, $n = 3$ biological replicates for E17.5; E17.5 experiments included a rabbit IgG control for the rabbit SS18 antibody).

**Urea denaturation.** To assess the relative stability of BAF:BRWD1 interactions, nuclear extracts were subjected to partial urea denaturation. Specifically, 300 µg of nuclear extract was resuspended in 100 µL of IP buffer (+0.5 mM CaCl₂) and then a freshly prepared 2× urea solution (in IP buffer) was added 1:1 for a final concentration of 0, 0.25, 0.5, 1, 2, or 4 M urea, respectively. The urea-nuclear extract solutions were vortexed for 1 s at medium ("4") speed to mix and then incubated at room temperature for 15 min. Each 200 µL solution was then dialyzed individually in a 3500 MWCO Slide-A-Lyzer MINI Dialysis Unit (ThermoFisher) with two 10 mL buffer exchanges over 2 h at 4 °C. The dialysis membrane was ruptured and all nuclear extract was recovered by centrifugation. IP buffer was added to a final volume of 1200 µL to achieve the optimal 0.25 mg/mL concentration of brain nuclear extract for each IP. Nuclear proteins were immunoprecipitated using α-SMARCA4 (BRG1 H-10 clone, Santa Cruz cat. #374197) or mouse IgG as a control, and then run on SDS-PAGE and immunoblotted as above.

**Antibodies.** Antibodies used for immunoprecipitation were: mouse IgG (Santa Cruz cat. # sc-2025), α-SMARCA4 (BRG1 H-10, mouse monoclonal, Santa Cruz cat. # sc-374197), α-SS18L1#1 (CREST M-15, goat polyclonal, Santa Cruz cat. # sc-50912), α-SS18L1#2 (CREST D-7, mouse monoclonal, Santa Cruz cat. # sc-515827), α-SS18 (SS18 D6I4Z, rabbit monoclonal, Cell Signaling cat. # 21792), (α-SMARCC2 (BAF170 E-6, mouse monoclonal, Santa Cruz cat. # sc-17838), α-SMARCB1 (INI1/BAF47 A-5, mouse monocolonal, Santa Cruz cat. # sc-166165), α-SMARCD3 (BAF60C RN-18, mouse monoclonal, Santa Cruz cat. # sc-101163; this antibody did not appear to work for IP). For immunoblotting, additional antibodies were used: α-HA (6E2, mouse monoclonal, Cell Signaling cat. # 2367; this antibody was by far the most clean of several α-HA antibodies tested), α-SMARCA2/4 (BRG1/BRM J1 clone, rabbit polyclonal, made in-house), α-ACTL6B/BAF53B (rabbit polyclonal, made in-house), α-ACTL6A/BAF53A (mouse monoclonal, NeuroMab clone N336B/83), α-TOP2B (F-12, rabbit polyclonal, Santa Cruz cat. # sc-365916), α-ARID1B (mouse monoclonal, Novus Biologicals cat. # H00057492-M01), α-PBRM1 (BAF180 D3F7O, rabbit monoclonal, Cell Signaling cat. #91894), α-TBP (mouse monoclonal, Abcam cat. # ab818), α-B-ACTIN (AC-15, mouse monoclonal, Santa Cruz cat. #sc-69879). Goat or donkey, α-mouse, or α-rabbit IRDye 800CW or 680LT (LI-COR) secondary antibodies were used for Western blot analysis with an Odyssey CLX imaging system (LI-COR).

**Western blot validations of tag expression in testes (related to Fig. S6).** Testes were harvested from 8-week old wildtype, heterozygous, and homozygous tagged mice and were homogenized in T-PER Protein Extraction Reagent (ThermoFisher Scientific) buffer with complete Protease Inhibitor Cocktail added (Sigma-Aldrich). Tissues were then centrifuged at 2900 RCF for 15 min at 4° to get rid of any pelleted debris, followed by sonication with 3 × 10 s pulses. Samples were then denatured in SDS loading buffer at 95° for 10 min and loaded into Mini-PROTEAN Pre-cast 4–20% Gradient Polyacrylamide Gels (BIO-RAD). Protein was transferred onto a nitrocellulose membrane at 1 amp for 40 min in Bolt & Mahoney buffer. The blot was blocked in 5% nonfat milk diluted in TBS with 0.1% Tween20 for 1 h and incubated in anti-HA antibody (1:500) at 4° for overnight (Roche, cat# 11867423001). Next day, the blot was washed with TBS-Tween20 (0.1%) and incubated with goat anti-rat IgG HRP antibody (1:5000) for 1 h at room temperature (abcam, cat# ab97057). To detect protein, blot was incubated with Luminata Crescendo Western HRP substrate (EMD Millipore) for 5 min and visualized using ChemiDoc Imaging System (BIO-RAD).

**Chromatin immunoprecipitation (CHIP)**
Hippocampal tissues (unilateral hippocampus from two animals were pooled/biological replicate; euploid, Ts65Dn, and Ts65Dn;*Brwd1*⁺/⁻, 6 wk old) were crosslinked, quenched and processed, as previously described[65]. Briefly, samples were washed thoroughly, lysed and sonicated. Samples were then incubated with a custom anti-SMARCA2/4 antibody provided by the Crabtree lab[66] (7.5 µg/sample) bound to sheep anti-rabbit M-280 Dynabeads (Invitrogen) on a rotator at 4 °C O/N. The following day, immunoprecipitates were washed, eluted and reverse-crosslinked in elution buffer O/N. DNA was purified using a PCR purification kit (Qiagen).

**CHIP-SEQ library preparation and sequencing**
Following DNA purifications, libraries were prepared using the TruSeq ChIP Library Prep Kit according to Illumina protocols, multiplexed library sizes were validated on an Agilent Bioanalyzer system and then sequenced (single-read) on an Illumina HiSeq 4000 sequencer.

**ATAC-SEQ preparation and sequencing**
**Nuclei isolation for (fluorescence-activated nuclear sorting) FANS.** Frozen whole hippocampus was homogenized in cold lysis buffer (0.32 M Sucrose, 5 mM CaCl₂, 3 mM Mg(Ace)2, 0.1 mM, EDTA, 10 mM Tris-HCl, pH8, 1 mM DTT, 0.1% Triton X-100) by douncing 50× in a 10 ml dounce homogenizer and filtered through a 40 µm cell strainer to isolate nuclei. The flow-through was underlaid with sucrose solution (1.8 M Sucrose, 3 mM Mg(Ace)2, 1 mM DTT, 10 mM Tris-HCl, pH8) and subjected to centrifugation at 4000 rpm for 15 min at 4 °C to remove cellular debris. Pellets were thoroughly resuspended in 500 µl DPBS and incubated in BSA (final concentration 0.1%) and anti-NeuN antibody (1:1000, Alexa488 conjugated, Millipore, cat#: MAB377X) under rotation for 1 h at 4 °C in the dark. Prior to FANS, DAPI (Sigma cat#: MBD0015) was added to a final concentration of 1 µg/ml. Unstained nuclei and nuclei stained with only secondary antibody served as negative controls. DAPI-positive neuronal (NeuN+) nuclei were sorted into tubes using a BD-FACSAria flow cytometer (BD Biosciences) equipped with a 70 µm nozzle.

**Generation of ATAC-seq libraries.** ATAC-seq reactions were performed using an established protocol[67] with minor modifications. Following FANS, 50,000 sorted nuclei were centrifuged at 500 × g for 10 min, 4 °C. Pellets were re-suspended in transposase reaction mix [25 µL 2× TD Buffer (Illumina Cat #FC-121-1030), 2.5 µL Tn5 Transposase (Illumina Cat #FC-121-1030) and 22.5 µL Nuclease Free H₂O] on ice. Samples were incubated at 37 °C for 30 min and then purified using the MinElute Reaction Cleanup kit (Qiagen Cat #28204) according to manufacturer's instructions. Following purification, library fragments were amplified using the Nextera index kit (Illumina Cat #FC-121-1011), under the following cycling conditions: 72 °C for 5 min, 98 °C for 30 s, followed by thermocycling at 98 °C for 10 s, 63 °C for 30 s and 72 °C for 1 min for a total of five cycles. In order to prevent saturation due to over-amplification, a 5 µl aliquot was then removed and subjected to qPCR for 20 cycles to calculate the optimal number of cycles needed for the remaining 45 µL reaction. The additional number of cycles was determined as follows: (1) Plot linear Rn vs. Cycle,(2) Calculate the # of cycle that is corresponded to 1/4 of maximum fluorescent intensity. In general, we found adding 4–6 cycles to this estimate yielded optimal ATAC-seq libraries. Libraries were amplified for a total of 13–19 cycles. Following PCR, ATAC-seq libraries were purified and double-end size-selected by (0.5× ratio and 1.8× ratio) Ampure XP bead purification (Beckman Coulter cat#; A63881) to remove primer-dimers (<100 bp) and large fragments (>1000 bp). Library size was quantified using Bioanalyzer High Sensitivity DNA Chips (Agilent technologies Cat#5067-4626). Libraries were quantified by quantitative PCR (KAPA Biosystems Cat#KK4873) prior to sequencing. Libraries were sequenced on NovaSeq 6000 System (Illumina) obtaining 2 ×50 paired-end reads.

## CHIP-SEQ/ATAC-SEQ analyses

For SMARCA2/4 ChIP-seq and neuronal ATAC-seq from adult mouse hippocampus, raw sequencing reads were aligned to the mouse genome (mm10) using default settings of HISAT2[62]. Only uniquely mapped reads were retained. Alignments were filtered using SAMtools (v1.19)[68] to remove duplicate reads. For ChIP-seq, peak-calling–normalized to respective inputs–was performed using MACS (v2.1.124)[69] with default settings; the window size was set as 300 bp. For ATAC-seq, peak calling was performed using MACS (v2.1.124) with settings --nomodel --shift −100 --extsize 200. For both ChIP-seq and ATAC-seq datasets, peaks were filtered for FDR < 0.05 and fold change >1.2. Differential analyses were performed using diffReps[70] with a window size of 1 kb. A default p-value cutoff of 0.0001 was used. Peaks and differential sites were further annotated to nearby genes or intergenic regions using the region analysis tool from the diffReps package (v1.55.6). Histone PTM enrichment data in mouse hippocampus were extracted from published sources[71]. ATAC-seq differential lists were compared to ChIP-seq differential lists based on gene names using geneOverlap (*GeneOverlap: Test and visualize gene overlaps*. R package version 1.23.0, http://shenlab-sinai.github.io/shenlab-sinai/), and heat maps were drawn using the deepTools package[72].

**Overlap with existing epigenomic annotations.** We computed the overlap of significant differentially bound sites of SMARCA2/4 (Ts65Dn *vs.* euploid) and chromatin states from the Epigenomics Roadmap Project[73,74] using the scaled Jaccard index, obtained by calculating standard deviations after subtracting the mean of the sample (Jaccard index is an intersection of base pairs divided by the union of base pairs). We used the "expanded" chromHMM 18-state (6 histone marks, 98 epigenome model) for seven brain regions, i.e., angular gyrus, anterior caudate, cingulate gyrus, dorsolateral prefrontal cortex, hippocampus, inferior temporal lobe and substantia nigra. To improve interpretability, we consolidated the 18 states into 9 states as follows: Promoter (TssA, TssFlnk, TssFlnkU, and TssFlnkD), Enhancer (EnhG1, EnhG2, EnhA1, EnhA2, EnhWk), Transcription (Tx, TxWk), Poised promoter (TssBiv), Repressed enhancer (EnhBiv), Repressed (ReprPC, ReprPCWk), Heterochromatin (Het), Repeats (ZNF/Rpts) and Low (Quies).

To compare our significant differentially bound sites with known regions of open chromatin in different brain regions, we used imputed versions of DNase-seq datasets from Epigenomics Roadmap Project.

## Statistical analysis

Statistical analyses were performed using Prism software (GraphPad). For all behavioral testing and molecular experiments involving more than two conditions, two-way or one-way ANOVAs were performed with subsequent *post hoc* analyses. For experiments comparing only two conditions, two-tailed Student's *t* tests were performed. In molecular analyses, all animals used were included as separate *n*s (i.e., samples were not pooled). Significance was determined at $p < 0.05$. All data are represented as mean ± SEM.

## Reporting summary

Further information on research design is available in the Nature Research Reporting Summary linked to this article.

## Data availability

Chromatin states determined by expanded 18-state ChromHMM model were downloaded from: http://egg2.wustl.edu/roadmap/data/byFileType/chromhmmSegmentations/ChmmModels/core_K27ac/jointModel/final/ DNAseq-seq dataset was downloaded from: https://egg2.wustl.edu/roadmap/data/byFileType/peaks/consolidated Imputed/narrowPeak/. Data from RNA-seq, ChIP-seq and ATAC-seq experiments have been deposited in the National Center for Biotechnology Information Gene Expression Omnibus (GEO) database

under accession numbers GSE210117 and GSE151255. We declare that the data supporting findings for this study are available within the article and Supplementary Information (see Supplemental Fig. 15 at the end of the Supplementary Information file for all uncropped blots with MW markers performed in this study). Source Data are provided with this paper. No restrictions on data availability apply. Source data are provided with this paper.

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

## Acknowledgements

We would like to thank Dr. William Mobley (UC San Diego School of Medicine) for expert advice on our work, Dr. Anita Bhattacharyya (University of Wisconsin) for providing us with DS hiPSCs, and Dr. Alexey Soshnev (The Rockefeller University) for help with illustrations. We would also like to acknowledge the Cornell University Stem Cell and Transgenic Core Facility for their help with CRISPR microinjections to generate *Brwd1*^{FLAG-HA} mice (special thanks to Robert Munroe and Christian Abratte). This work was supported by grants from the National Institutes of Health: R01 HD097088 (I.M.), F99 NS125774 (S.L.F.), F31 MH116588 and F99 NS118735 (W.W.), R15 HD090603 (R.J.R.), R01 HD082568 (J.C.S.), SC2 GM122646 (A.K.F.), R01 MH111604 (A.J.R.), R01 NS085171 (A.J.R.), R01 DA040621 (A.J.R.), R01 CA163915 (G.R.C.), and R37 NS046789 (G.R.C.), as well as awards from: March of Dimes (I.M.), Alfred P. Sloan Foundation Fellowship in Neuroscience (I.M.), Avielle Foundation (A.J.R.), CIRM RB4-05886 (G.R.C.), SFARI (G.R.C.) and the Howard Hughes Medical Institute (I.M. and G.R.C.).

## Author contributions

I.M. conceived of the project with input from S.L.F. A.E.L., and W.W. S.L.F, W.W., A.E.L., A.L.E., T.F., R.M.B., E.C.H., A.N., L.A.F., A.A-K., Y.L., B.C., T.N.T., K.J.B., J.C.S., A.K.F., R.D.B., A.J.R., G.R.C., and I.M. designed and/or executed the experiments and interpreted the data. A.R., S.L.F., J.B., J.C.C., P.R., and L.S. performed the sequencing-based bioinformatics with input from A.E.L. and I.M. R.L.N. generated the HSV vectors. R.J.R. generated relevant brain tissues from transgenic mice for molecular analyses. S.L.F, W.W., A.E.L., and I.M. wrote the manuscript.

## Competing interests

The authors declare no competing interests.

## Additional information

¹Nash Family Department of Neuroscience, Friedman Brain Institute, Icahn School of Medicine at Mount Sinai, New York, NY 10029, USA. ²Department of Pathology, Stanford Medical School, Palo Alto, CA 94305, USA. ³Department of Genetics, Stanford Medical School, Palo Alto, CA 94305, USA. ⁴Department of Developmental Biology, Stanford Medical School, Palo Alto, CA 94305, USA. ⁵Howard Hughes Medical Institute, Stanford University, Palo Alto, CA 94305, USA. ⁶Department of Physiology, Michigan State University, East Lansing, MI 48824, USA. ⁷Department of Pharmacological Sciences, Icahn School of Medicine at Mount Sinai, New York, NY 10029, USA. ⁸Department of Psychiatry, Icahn School of Medicine at Mount Sinai, New York, NY 10029, USA. ⁹Department of Genetics and Genomics, Icahn School of Medicine at Mount Sinai, New York, NY 10029, USA. ¹⁰Center for Disease Neuroepigenomics, Icahn School of Medicine at Mount Sinai, New York, NY 10029, USA. ¹¹Department of Biomedical Sciences, Cornell University, Ithaca, NY 14853, USA. ¹²Department of Molecular Biology and Genetics, Cornell University, Ithaca, NY 14853, USA. ¹³McGovern Institute for Brain Research, Massachusetts Institute of Technology, Cambridge, MA 02139, USA. ¹⁴Department of Biology, Indiana University-Purdue University, Indianapolis, IN 46202, USA. ¹⁵Black Family Stem Cell Institute,

Icahn School of Medicine at Mount Sinai, New York, NY 10029, USA. [16]J.J. Peters Veterans Affairs Hospital, Bronx, NY 10468, USA. [17]Department of Biological Sciences, City University of New York-Hunter College, New York, NY 10065, USA. [18]Howard Hughes Medical Institute, Icahn School of Medicine at Mount Sinai, New York, NY 10029, USA. [19]Present address: Departments of Psychiatry and Genetics, Wu Tsai Institute, Yale School of Medicine, New Haven, CT 065109, USA. [20]These authors contributed equally: Sasha L. Fulton, Wendy Wenderski, Ashley E. Lepack. ✉e-mail: ian.maze@mssm.edu

