## [Peer Review File · Nature Communications]

REVIEWER COMMENTS

Reviewer #1 (Remarks to the Author):

This paper demonstrates that the chromatin effector Brwd1 is aberrantly elevated in Ts65Dn mice which phenocopy Down Syndrome. Genetic restoration of Brwd1 copy number rescues synaptic and cognitive defects in male mice. Additionally, targeted elevation of Brwd1 in the hippocampus enhanced neuron excitability and resulted in a loss of immediate early gene activation. Mechanistically, the authors demonstrate Brwd1 associates with the BAF complex and Brwd1 elevation may contribute to BAF mistargeting, dysregulated chromatin structure and alterations to gene expression which could be partially rescued with Brwd1 copy number restoration. The findings presented in this manuscript are novel, exciting and will be of great interest to several groups including the Down Syndrome community. However, there is an obvious sex difference that is not discussed. Furthermore, the authors do not definitively demonstrate that Brwd1 is associated with nBAF and not npBAF nor do they provide sufficient evidence to demonstrate nBAF is retargeted to inactive chromatin with the elevation of Brwd1.

Major Points:

1. A major issue with this study is the very extreme sex difference observed with respect to Brwd1 over-expression contributing to behavioural deficits in the Down Syndrome mouse model. Male Ts65Dn mice exhibit defects in LTP as well as FC memory which can be rescued following Brwd1 copy number restoration. Female mice have no differences in LTP but have deficits in FC memory that isn't rescued with Brwd1 copy number restoration. It is unclear where this sex difference is coming from. Is Brwd1 over expressed in the Ts65Dn female mice? Given the very different phenotypes present in male vs female Ts65Dn mice, Brwd1 mRNA expression (Figure 1a) should be separated by sex for both the E17.5 and 6 week timepoints. Additional discussion as to why males and females show unique responses and relevance to human Down Syndrome should also be included.
2. The evidence for Brwd1 being a dedicated nBAF subunit is incomplete. Although BRWD1-FLAG-HA was depleted from nuclear extracts with an antibody to the neuronal-specific subunit SS18L1 the authors should determine if BRWD1 is incorporated into npBAF. This is of particular interest because Brwd1 copy number rescued 60% of the aberrant gene expression at E17.5 compared to 19.2% in 6-week-old males. For example, it would be helpful to probe the co-IP blots for ACTL6A to determine exactly which complexes Brwd1 is found.
3. The authors present some evidence to suggest BAF may be retargeted to repressive chromatin, however, they do not directly demonstrate this. Figure 4a is also confusing in the way it is displayed. The histone marks for permissive versus repressive chromatin are supposed to suggest that the Ts65Dn SMARCA2/4 peaks are enriched over repressed regions in normal mice? What statistics were conducted for this type of enrichment to see if this enrichment is greater than expected by chance? This also doesn't necessarily mean that the peaks for SMARCA2/4 in Ts65Dn are repressed. The authors should perform ChIP-qPCR for SMARCA2/4 and the repressive marks at several examples of these sites to determine if SMARCA2/4 is actually present at areas with repressive marks in Ts65Dn mice. Similarly, validation by ChIP-qPCR of sites that are then rescued in the Ts65Dn-Brwd1^{+/-} mice would also be necessary to conclude that mistargeting to repressive chromatin is rescued in the double mutant.

Minor Points:

1. Lines 148-149: "BRWD1 is found in three copies in Ts65Dn mice and in individuals with DS" should be appropriately referenced.
2. For graphs depicting mRNA expression, what is the definition of A.U.? is mRNA expression

relative to a housekeeping gene?

3. Can the authors discuss why *Brwd1*^{+/-} mice have half the *Brwd1* mRNA expression of euploid controls but *Ts65Dn* mice only have 1.5-2 fold increase in expression? Would it not be expected that *Brwd1* mRNA levels be 3 fold that of euploid controls?
4. Can the authors elaborate on what evidence they provide to support lines 276-277: "Given that the majority of BRWD1-FLAG-HA interacts with BAF (Fig. 3e-f), this suggests that BRWD1-containing BAF complexes represent only a fraction of the total BAF pool [which is ~300,000 complexes per neuron]"?
5. Extended data figure 3 states "Swim distance during training (l) and during the probe test (m), as well as swim speed during training (n) and during the probe test (o) were unaffected ($p < 0.05$ for all comparisons)". In particular, swim speed (n) appears to be negatively affected in HSV-BRWD1-GFP mice on days 3-5. Do these mice have alterations to swim speed or is this an error in statistic reporting?
6. Figure 3f is unclear as to how many biological replicates are represented. Please include individual data points and error bars.
7. The authors should include more information on how the RNA-seq and ChIP-seq libraries were prepared and sequenced. Please include quality control measures, kits used, sequencing conditions (paired or single end) and sequencer at minimum.

Reviewer #2 (Remarks to the Author):

In this manuscript the authors examine the gene *Brwd1* as a candidate for the neurological impairments that arise in Down's Syndrome from trisomy of chromosome 21. Multiple genes are found in the interval that is triplicated in Down Syndrome, and of those genes the functions of *Brwd1* are poorly understood. The authors use the *Ts65Dn* model of trisomy 21 which replicates the increased copy number of some of these genes. They validate what was reported elsewhere, which is that *Brwd1* expression is elevated in this model, and then the cross to a *Brwd1* HET, which they show reduces *Brwd1* expression. Males *Ts65Dn* mice show LTP, behavioral, and gene expression deficits that are improved with *Brwd1* copy number resolution (females show a more complex and variable phenotype). The most interesting part of the story comes when the authors ask what function *Brwd1* might perform. They make a tagged knockin mouse and use the tag to purify *Brwd1* on a glycerol gradient in a large multiprotein complex. They appear to make a good guess based on the properties of the *Brwd1* complex that it may be interacting with BAF and do a series of experiments to show that nBAF is mistargeted in *Ts65Dn* brain.

Overall this is a novel series of findings that enhance understanding of an important and understudied neurodevelopmental disorder. I have only minor concerns.

- 1) Why is the fold enrichment of *Brwd1* mRNA in figure 1A about 1.5 fold but in figure 1C it is 2.5 fold. This matters because the *Brwd1* HET brings the amount back to about 1.5 fold, which is a rescue in panel 1C but not in 1A.
- 2) What do *Brwd1* levels look like in male versus female mice (control and *Ts65Dn*) given the differences in behavior, LTP, gene expression, and rescue?
- 3) Do the authors have suggested reasons for the sex differences? Are there sex differences in the expression of Down Syndrome phenotypes in humans?

4) The HSV appears to infect a very small percentage of the cells in hippocampus in extended data 3, which means the 1.5fold average overexpression in a tissue punch is likely much higher on a per cell basis. I am guessing the authors used HSV instead of AAV because Brwd1 is very large? Nonetheless, it is not clear how relevant these data really are for comparison to the levels of Brwd1 overexpression in the trisomy model.

5) The paper could use a discussion that would talk about sex differences, potential mechanisms of nBAF recruitment, the plusses and minuses of the Ts65Dn model etc.

Reviewer #3 (Remarks to the Author):

This MS provides initial evidence for the involvement of the BRWD1 gene dosage in the phenotypic spectrum of Down syndrome.

The hypothesis is that because BRWD1 maps on chromosome 21 and it is overexpressed (as expected) in trisomy 21, AND it acts as a chromatin regulator, it is a candidate gene for contributing to certain phenotypes of Down syndrome. The candidacy of BRWD1 for trisomy 21 contributing gene was mentioned in PMID 24740065 (Letourneau et al Nature vol 508 pg 349). Thus I suggest that the authors include this paper in the introduction somewhere in the paragraph starting on line 140.

Concerns/Criticisms

1. The study is entirely done on the mouse model Ts65Dn. Although this model has been widely used, it also has disadvantages since it has a partial trisomy MMU16 AND a partial trisomy MMU17 that complicates the relevance to human trisomy 21.

2. Mouse Ts65Dn also contains 3 copies of the DYRK1A gene that several studies have shown its importance in the hippocampal FTP and other functional tests. Thus one needs to clarify/explain/discuss why the Ts65Dn with 3x Dyrk1a and 2x Brwd1 is "normal".

3. Since Ts65Dn is not the perfect model for trisomy 21, and many claims (including therapeutic ones) based on this model did not correspond to what is seen in humans, I suggest the authors to also perform some additional experiments in human trisomy 21 cells. For example inactivate one BRWD1 allele in a trisomy cell line by allelic CRISPR for example, and look at the differences in the transcriptome, or differences in neuronal cells after differentiation. Without a validation in a human cellular system, this study will be "yet one more Ts65Dn"... In contrast, the human experiment will make this study an important contribution to the DS research.

4. I found the model on extended data figure 10 rather weak; why for example the protein loaded with the 3 yellow circles (BAF complex) works in euploid brain and not in the Ts65Dn (top and bottom of the scheme)?

5. Yansheng Liu in PMID: 29089484 (NCOMM 2017) has shown that proteins in complexes are well buffered in trisomy 21 and their amount is not increased as one might expect from the RNA data. Wondering if the authors could measure the BRWD1 protein in the mouse hippocampal extracts.

I found the exploration of the role of BRWD1 of great importance in the Down syndrome field, and thus this MS provides an initial and serious step in this direction.

Response to Referees

Reviewer #1:

This paper demonstrates that the chromatin effector Brwd1 is aberrantly elevated in Ts65Dn mice, which phenocopy Down Syndrome. Genetic restoration of Brwd1 copy number rescues synaptic and cognitive defects in male mice. Additionally, targeted elevation of Brwd1 in the hippocampus enhanced neuron excitability and resulted in a loss of immediate early gene activation. Mechanistically, the authors demonstrate that Brwd1 associates with the BAF complex, and Brwd1 elevation may contribute to BAF mistargeting, dysregulated chromatin structure and alterations to gene expression, which could be partially rescued with Brwd1 copy number restoration. The findings presented in this manuscript are novel, exciting and will be of great interest to several groups including the Down Syndrome community. However, there is an obvious sex difference that is not discussed. Furthermore, the authors do not definitively demonstrate that Brwd1 is associated with nBAF and not npBAF, nor do they provide sufficient evidence to demonstrate nBAF is retargeted to inactive chromatin with the elevation of Brwd1.

Response: We very much thank the Reviewer for commenting that our manuscript is “novel, exciting and will be of great interest to several groups including the Down Syndrome community.” As discussed below, we have added substantial new data and discussions to this Resubmission in order to provide greater context for the sex differences observed in the trisomic mouse model, as well as for Brwd1-BAF complex interactions and roles for Brwd1 in genomically mis-targeting the BAF complex in trisomic brain. Please find our detailed responses to the concerns raised below.

Major Points:

1. A major issue with this study is the very extreme sex difference observed with respect to Brwd1 over-expression contributing to behavioural deficits in the Down Syndrome mouse model. Male Ts65Dn mice exhibit defects in LTP as well as FC memory, which can be rescued following Brwd1 copy number restoration. Female mice have no differences in LTP but have deficits in FC memory that isn't rescued with Brwd1 copy number restoration. It is unclear where this sex difference is coming from.

Is Brwd1 over expressed in the Ts65Dn female mice? Given the very different phenotypes present in male vs. female Ts65Dn mice, Brwd1 mRNA expression (Figure 1a) should be separated by sex for both the E17.5 and 6-week time points. Additional discussion as to why males and females show unique responses and relevance to human Down Syndrome should also be included.

Response: We fully agree that the observed sex differences are intriguing, which is why we still include data from both sexes in the revised manuscript (especially given that very few studies to date have assessed phenotypes in female Ts65Dn mice). In the revised version of our paper, we provide additional qPCR data comparing *Brwd1* expression in male vs. female hippocampus, as well as in euploid Ts65Dn brain at both E17.5 and 6-

weeks of age (Fig. 1a – male, and Extended Data Fig. 2b – female). We found that *Brwd1* mRNA is elevated in brain of trisomic animals regardless of sex, both at E17.5 and 6-weeks of age.

These data complement our RNA-seq results, which also demonstrate that *Brwd1* expression is elevated in both trisomic males and females. And importantly, the Ts65Dn;*Brwd1*^{+/-} cross selectively rescued *Brwd1* triplication in Ts65Dn mice (Fig. 1c) without directly genetically restoring the trisomic background. However, in female mice, while *Brwd1* expression was significantly reduced in Ts65Dn;*Brwd1*^{+/-} vs. Ts65dn animals, we identified only trending *Brwd1* increases in Ts65Dn vs. euploid mice when comparing all four genotypes (Extended Fig. 2c), perhaps reflecting higher variability in *Brwd1* upregulation in female Ts65Dn hippocampus. Indeed, female Ts65Dn mice displayed more modest hippocampal *Brwd1* fold-change (FC) differences vs. euploid (average 1.19 FC) in comparison to those detected in males (average 1.41 FC). These findings are consistent with previous studies in which BRWD1 has been shown to display sex-specific functions and expression levels, particularly in the context of early reproductive cellular genesis.

Similarly, although we did not detect altered hippocampal LTP in female Ts65Dn mice (Extended Data Fig. 2d), we did find that female Ts65Dn mice scored worse in the contextual FC memory task vs. euploid controls (Extended Data Fig. 2e), suggesting that distinct molecular mechanisms may contribute to contextual fear learning in male vs. female Ts65Dn mice. Furthermore, *Brwd1* copy number restoration did not significantly rescue these cognitive deficits, reflecting a more limited contribution of *Brwd1* to Ts65Dn hippocampal function in females.

Finally, in female trisomic mice (vs. their male counterparts), a more modest rescue (~9.2%) of hippocampal gene expression changes with *Brwd1* normalization were observed (Extended Data Fig. 8a-b), and associated processes/pathways for rescued genes in female mice were distinct from those seen in males, with significant GO term enrichment identified for protein synthesis and translation, as well as neuronal differentiation (Extended Data Fig. 8c). In addition, genes that were found to be differentially expressed between female vs. male Ts65Dn hippocampus most significantly associated with LTP and neuronal morphology, highlighting potential molecular and/or anatomical differences between the sexes in the pathophysiology of DS-related deficits (Extended Data Fig. 8d). Notably, in all cases (E17.5 forebrain, adult male and female hippocampus), differentially expressed genes from Ts65Dn animals were not limited to trisomic loci, consistent with previous findings, and *Brwd1* renormalization reversed the expression of many of these genes across all chromosomes (Extended Data Fig. 9a).

We have now added an extended Discussion section to the manuscript, where this issue of sex differences is more fully elaborated upon, including further discussions on the potential relevance of our findings to human Down syndrome.

2. The evidence for Brwd1 being a dedicated nBAF subunit is incomplete. Although BRWD1-FLAG-HA was depleted from nuclear extracts with an antibody to the neuronal-

specific subunit SS18L1, the authors should determine if BRWD1 is incorporated into npBAF. This is of particular interest because Brwd1 copy number rescued 60% of the aberrant gene expression at E17.5 compared to 19.2% in 6-week-old males. For example, it would be helpful to probe the co-IP blots for ACTL6A to determine exactly which complexes Brwd1 is found.

Response: The Reviewer raises an excellent question as to whether BRWD1 interacts with BAF during embryonic brain development. As highlighted by the Reviewer, neural progenitor BAF complexes contain SS18, ACTL6A (BAF53a) and DPF2 (BAF45d), which are replaced with homologous subunits SS18L1 (CREST), ACTL6B (BAF53b) and DPF1/3 (BAF45b/c) in neurons. At E17.5, the brain contains both neural progenitors and immature neurons that prominently express npBAF or nBAF subunits, respectively. To learn if BRWD1 associates with npBAF and/or nBAF in embryonic brain, we immunoprecipitated BAF complexes from E17.5 brain nuclear extracts with antibodies to SS18 for npBAF, SS18L1 for nBAF, SMARCA4 for all BAF complexes, or IgG as a control. In embryonic brain, we found that BRWD1 strongly associated with the core BAF subunit SMARCA4 and the npBAF subunit SS18, and to a lesser degree with the nBAF subunit SS18L1. To further characterize this interaction, we subjected E17.5 brain nuclear extracts to 3 rounds of immunodepletion with these antibodies and found that BRWD1 was significantly co-immunodepleted with SS18 by ~20%. From adult brain nuclear extracts, a C-terminal SS18L1 antibody (#1) was able to deplete BRWD1 by ~80% after 3 rounds of immunodepletion, while an antibody raised against an internal peptide of SS18L1 (#2) immunodepleted BRWD1 by ~10% and an antibody to SMARCA4 immunodepleted BRWD1 by ~20%. Notably, these antibodies selectively and near-completely depleted their target proteins but immunodepleted non-target BAF subunits by only 20-50%. While BRWD1 and BAF subunits behaved similarly in these assays, we could not determine if BRWD1 was a dedicated BAF subunit because neither BAF complexes or BRWD1 could be fully immunodepleted with the antibodies tested. We now make this clear in the text and conclude that BRWD1 associates with BAF complexes in both embryonic and adult brain. These data are now provided in Fig. 3 and Extended Data Figs. 11-12. We have also added an extended Discussion section to the manuscript, where we now discuss the implications of Brwd1-BAF interactions in greater detail and posit that Brwd1 likely exists as a sub-stoichiometric, yet critical, component of all canonical BAF complexes in developing and adult brain.

3. The authors present some evidence to suggest BAF may be retargeted to repressive chromatin, however, they do not directly demonstrate this. Figure 4a is also confusing in the way it is displayed. The histone marks for permissive versus repressive chromatin are supposed to suggest that the Ts65Dn SMARCA2/4 peaks are enriched over repressed regions in normal mice? What statistics were conducted for this type of enrichment to see if this enrichment is greater than expected by chance? This also doesn't necessarily mean that the peaks for SMARCA2/4 in Ts65Dn are repressed. The authors should perform ChIP-qPCR for SMARCA2/4 and the repressive marks at several examples of these sites to determine if SMARCA2/4 is actually present at areas with repressive marks in Ts65Dn mice. Similarly, validation by ChIP-qPCR of sites that are then rescued in the Ts65Dn-

Brwd1^{+/-} mice would also be necessary to conclude that mistargeting to repressive chromatin is rescued in the double mutant.

Response: The Reviewer is correct that the heat map comparisons (now in Extended Data Fig. 13b) against previously published histone mark genomic enrichment profiles in adult hippocampus indicated that SMARCA2/4 is normally enriched at permissive loci (primarily at promoters and putative enhancers) in euploid animals and becomes mistargeted in Ts65Dn hippocampus towards intergenic regions of the genome that are depleted of active PTMs in wildtype animals. We also provide a more definitive demonstration that sites of aberrant SMARCA2/4 enrichment in trisomic animals map strongly onto well annotated, repressed regions of the genome in human brain (see Fig. 4c and Extended Data Fig. 14a-b). These comparisons against the epigenomically annotated human brain demonstrate that in Ts65Dn hippocampus, SMARCA2/4 is mistargeted away from promoters (both poised and active) and inappropriately enriched throughout regions of the genome classified as being “repressed,” “heterochromatic,” “repetitive” and “low” in expression (details on these classifications can be found in the Methods section of the manuscript). Interestingly, further odds ratio analyses revealed that PCGs that lost SMARCA2/4 binding in Ts65Dn vs. euploid hippocampus overlapped most significantly with upregulated genes in both Ts65Dn and human DS, as well as with rescued downregulated genes in Ts65Dn;*Brwd1*^{+/-} mice, suggesting that BAF complex mistargeting away from “active” promoters is associated with inappropriate induction of gene expression (Extended Data Fig. 13d-e). In agreement with these data, we identified a significant association (~85%, 506/595) between BRWD1-rescued PCGs in both SMARCA2/4 enrichment and chromatin accessibility, indicating that BRWD1-mediated BAF mistargeting is also significantly associated with dysregulated chromatin structure (Fig. 4f).

In accordance with the Reviewer’s suggestion, we additionally performed ChIP-seq for H3K27me3 in adult male hippocampus from euploid vs. Ts65Dn vs. Ts65Dn;*Brwd1*^{+/-} mice, and while subtle correlations were identified between loci showing aberrant trisomy induced enrichment of SMARCA2/4 and repressive H3K27me3, such overlaps did not fully encompass the full repertoire of SMARCA2/4 bound loci in Ts65Dn brain (these data are not included in the current resubmission). As such, the question of whether BRWD1 directly binds to specific histone modifications in brain to mediate BAF (mis)targeting, or to which histone marks it may bind, remains to be determined in future studies. Given this, we have removed comparative heatmaps against repressive histone PTMs (e.g., H3K27me3 and H3K9me3) in Extended Fig. 13b and have removed assertions in the manuscript regarding which histone PTMs, if any, *Brwd1* may target BAF towards in trisomic brain. We instead focus our discussions on BAF complex depletion from permissive gene promoters and putative enhancers in trisomic brain and provide evidence that BAF mistargeting from these loci strongly correlate with altered chromatin accessibility and gene expression, both of which can be partly rescued with *Brwd1* renormalization.

In our resubmission, we have greatly improved descriptions of the data presented (and associated methodologies) and provide further details on the statistics used in these studies to improve clarity.

Minor Points:

1. Lines 148-149: “*BRWD1* is found in three copies in *Ts65Dn* mice and in individuals with *DS*” should be appropriately referenced.

Response: We have included appropriate references for this statement in the revised manuscript.

2. For graphs depicting mRNA expression, what is the definition of A.U.? is mRNA expression relative to a housekeeping gene?

Response: In this case, A.U. stands for Arbitrary Units, as the data are normalized to the euploid control group in order to display fold changes in *Brwd1* expression over that of wildtype animals. This is now explicitly defined in the Legend for Fig. 1. In addition, all mRNA expression data were normalized to *Gapdh*, a common housekeeping gene. We include this information in the revised manuscript in the Methods Section under “RNA isolation and qPCR.”

3. Can the authors discuss why *Brwd1*^{+/-} mice have half the *Brwd1* mRNA expression of euploid controls but *Ts65Dn* mice only have 1.5-2 fold increase in expression? Would it not be expected that *Brwd1* mRNA levels be 3 fold that of euploid controls?

Response: With the addition of a third copy of *Brwd1* in *Ts65Dn* animals, one would expect that *Brwd1* expression should be overexpressed by 1.5 fold vs. euploid mice (assuming that dosage compensation does not occur), which is what we observe – see Fig. 1a and 1c, as well as Extended Data Figs. 2a-b. Such fold change differences were confirmed in RNA-seq data from embryonic brain, cultured primary neurons, and adult mouse brain (both male and female) comparing *Ts65Dn* vs. euploid animals – all data sets which now include additional n for verification purposes. Given that the data are presented as a relative fold difference vs. euploid controls (which are set to an A.U. = 1), one would expect that in diploid animals, each copy of *Brwd1* would contribute equally to the A.U. = 1 (i.e., each copy contributes 0.5 of the total qPCR signal). Thus, with an additional copy of *Brwd1* in trisomic animals, the relative fold difference vs. euploid controls should increase by a factor of one (i.e., + 0.5 fold) for a difference of 1.5 fold.

4. Can the authors elaborate on what evidence they provide to support lines 276-277: “Given that the majority of *BRWD1*-FLAG-*HA* interacts with *BAF* (Fig. 3e-f), this suggests that *BRWD1*-containing *BAF* complexes represent only a fraction of the total *BAF* pool [which is ~300,000 complexes per neuron]”.

Response: We have now added further discussion of this critical point to the Discussion section of the manuscript. In effect, we hypothesize that *BRWD1* may act as a

substoichiometric component for a subset of BAF complexes in brain, in that it functions as a histone-targeting protein for a certain fraction of total BAF (including both npBAF and nBAF) in a context-specific manner. Its substoichiometric relationship to BAF is supported by its dosage-dependent effects on BAF targeting, the fact that *Brwd1* deletion does not affect BAF assembly or mass on a glycerol gradient, and BRWD1's lower protein expression in mouse brain relative to other BAF subunits. However, as discussed above, the observation that BAF was targeted away from "active" histone marks in trisomy does not necessarily mean that repressed regions are the primary target of BRWD1 in euploid neurons. In fact, BRWD1's bromodomain 2 was found to bind to transcriptionally-activating H3K14 and K18 acetylation in vitro, and its WD-repeat domain shares >84% identity with that of WDR5, which "reads" H3K4 as part of the MLL complex. Whether BRWD1 directly binds histone modifications in brain, which histone marks it may bind to, and if its putative histone binding activity may contribute to BAF mistargeting in trisomy remain to be determined.

5. *Extended data figure 3 states "Swim distance during training (l) and during the probe test (m), as well as swim speed during training (n) and during the probe test (o) were unaffected (p<0.05 for all comparisons)". In particular, swim speed (n) appears to be negatively affected in HSV-BRWD1-GFP mice on days 3-5. Do these mice have alterations to swim speed or is this an error in statistic reporting?*

Response: Excellent catch! This was an error in statistical reporting, and we thank the Reviewer for noticing this. It should have read as "p>0.05 for all comparisons." This has now been corrected in the text.

6. *Figure 3f is unclear as to how many biological replicates are represented. Please include individual data points and error bars.*

Response: In our revised manuscript, individual data points for BAF immunopurifications are now plotted where possible and/or relevant Ns are now explicitly stated in the figure legend (see Fig. 3 and Extended Data Fig. 12).

7. *The authors should include more information on how the RNA-seq and ChIP-seq libraries were prepared and sequenced. Please include quality control measures, kits used, sequencing conditions (paired or single end) and sequencer at minimum.*

Response: Further information on how RNA- and ChIP-seq libraries were prepared and sequenced is now included in the Methods section of our revised manuscript.

Reviewer #2

*In this manuscript the authors examine the gene *Brwd1* as a candidate for the neurological impairments that arise in Down's Syndrome from trisomy of chromosome 21. Multiple genes are found in the interval that is triplicated in Down Syndrome, and of those genes the functions of *Brwd1* are poorly understood. The authors use the *Ts65Dn**

model of trisomy 21, which replicates the increased copy number of some of these genes. They validate what was reported elsewhere, which is that Brwd1 expression is elevated in this model, and then the cross to a Brwd1 HET, which they show reduces Brwd1 expression. Males Ts65Dn mice show LTP, behavioral, and gene expression deficits that are improved with Brwd1 copy number resolution (females show a more complex and variable phenotype). The most interesting part of the story comes when the authors ask what function Brwd1 might perform. They make a tagged knockin mouse and use the tag to purify Brwd1 on a glycerol gradient in a large multiprotein complex. They appear to make a good guess based on the properties of the Brwd1 complex that it may be interacting with BAF and do a series of experiments to show that nBAF is mistargeted in Ts65Dn brain.

Overall this is a novel series of findings that enhance understanding of an important and understudied neurodevelopmental disorder. I have only minor concerns.

Response: We greatly appreciate this Reviewer's enthusiasm regarding our manuscript and their feeling that our findings are novel and important for the field. Please find our detailed responses to the "minor" concerns raised below.

1) Why is the fold enrichment of Brwd1 mRNA in figure 1A about 1.5 fold but in figure 1C it is 2.5 fold. This matters because the Brwd1 HET brings the amount back to about 1.5 fold, which is a rescue in panel 1C but not in 1A.

Response: With the addition of more animals to the comparisons displayed in Fig. 1a and 1c, we demonstrate that Brwd1 expression is elevated by ~1.5 fold in male trisomic vs. euploid animals (see Reviewer #1, Major Point 1 for a discussion of results in females; Extended Data Fig. 2a-b), an effect that is completely renormalized in Ts65Dn;Brwd1^{+/-} mice. Thus, these data are consistent with our findings in Fig. 1a.

2) What do Brwd1 levels look like in male versus female mice (control and Ts65Dn) given the differences in behavior, LTP, gene expression, and rescue?

Response: As noted also in response to Reviewer #1, in the revised version of our paper, we provide additional qPCR data comparing Brwd1 expression in male vs. female, euploid vs. Ts65Dn brain at both E17.5 and 6-weeks of age (Fig. 1a – male, and Extended Data Fig. 2b – female). We found that Brwd1 mRNA is elevated in brain of trisomic animals regardless of sex, both at E17.5 and 6-weeks of age. These data complement our RNA-seq results, which also demonstrate elevations in *Brwd1* expression in both trisomic males and females. And importantly, the Ts65Dn;Brwd1^{+/-} cross selectively rescued *Brwd1* triplication in male Ts65Dn mice without directly genetically restoring the trisomic background (Fig. 1c). However, in female mice, while *Brwd1* expression was significantly reduced in Ts65Dn;Brwd1^{+/-} vs. Ts65dn animals, we identified only trending *Brwd1* increases in Ts65Dn vs. euploid mice when comparing all four genotypes (Extended Fig. 2b), perhaps reflecting higher variability in *Brwd1* upregulation in female Ts65Dn hippocampus. Indeed, female Ts65Dn mice displayed more modest hippocampal *Brwd1* fold-change (FC) differences vs. euploid (average 1.19

FC) in comparison to those detected in males (average 1.41 FC). These findings are consistent with previous studies in which BRWD1 has been shown to display sex-specific functions and expression levels, particularly in the context of early reproductive cellular genesis.

Although we did not detect altered hippocampal LTP in female Ts65Dn mice (Extended Data Fig. 2d), as noted by the Reviewer, we did find that female Ts65Dn mice scored worse in the contextual FC memory task *vs.* euploid controls (Extended Data Fig. 2e), suggesting that distinct molecular mechanisms may contribute to contextual fear learning in male *vs.* female Ts65Dn mice. Furthermore, *Brwd1* copy number restoration did not significantly rescue these cognitive deficits, reflecting a more limited contribution of *Brwd1* to Ts65Dn hippocampal function in females.

Finally, in female trisomic mice (*vs.* their male counterparts), a more modest rescue (~9.2%) of hippocampal gene expression changes with *Brwd1* normalization were observed (Extended Data Fig. 8a-b), and associated processes/pathways for rescued genes in female mice were distinct from those seen in males, with significant GO term enrichment identified for protein synthesis and translation, as well as neuronal differentiation (Extended Data Fig. 8c). In addition, genes that were found to be differentially expressed between female *vs.* male Ts65Dn hippocampus most significantly associated with LTP and neuronal morphology, highlighting potential molecular and/or anatomical differences between the sexes in the pathophysiology of DS-related deficits (Extended Data Fig. 8d). Notably, in all cases (E17.5 forebrain, adult male and female hippocampus), differentially expressed genes from Ts65Dn animals were not limited to trisomic loci, consistent with previous findings, and *Brwd1* renormalization reversed the expression of many of these genes across all chromosomes (Extended Data Fig. 9a).

We have now added an extended Discussion section to the manuscript, where this issue of sex differences is more fully elaborated upon, including further discussions on the potential relevance of our findings to human Down syndrome.

3) Do the authors have suggested reasons for the sex differences? Are there sex differences in the expression of Down Syndrome phenotypes in humans?

Response: Please see response to point #2 above. In addition, we have added an extended Discussion section to the revised manuscript, where this issue of sex differences is more fully elaborated upon, including further discussions on the potential relevance of our findings to human Down syndrome.

*4) The HSV appears to infect a very small percentage of the cells in hippocampus in extended data 3, which means the 1.5 fold average overexpression in a tissue punch is likely much higher on a per cell basis. I am guessing the authors used HSV instead of AAV because *Brwd1* is very large? Nonetheless, it is not clear how relevant these data really are for comparison to the levels of *Brwd1* overexpression in the trisomy model.*

Response: The Reviewer is correct that HSV vectors were required in this case given the large size of the *Brwd1* insert, which far exceeds the capacity of traditional AAV vectors, and to ensure neuronal specific expression. These data are presented to provide further evidence that *Brwd1* overexpression, outside of the trisomic milieu, is sufficient to promote cognitive deficits that are commonly observed in DS. Having said this, we agree that these experiments are not directly comparable to the transgenic mouse studies, and as such, we have now modified our interpretation of these results in this section of the manuscript so as not to imply that they are directly comparable.

5) The paper could use a discussion that would talk about sex differences, potential mechanisms of nBAF recruitment, the plusses and minuses of the Ts65Dn model etc.

Response: As mentioned in points #2 and 3 above, we have now added an extended Discussion section to the manuscript, where the issues of both sex differences and the Ts65Dn model have been more fully elaborated upon, including further discussions on the potential relevance of our findings to human Down syndrome.

Reviewer #3

This MS provides initial evidence for the involvement of the BRWD1 gene dosage in the phenotypic spectrum of Down syndrome. The hypothesis is that because BRWD1 maps on chromosome 21 and it is overexpressed (as expected) in trisomy 21, AND it acts as a chromatin regulator, it is a candidate gene for contributing to certain phenotypes of Down syndrome. The candidacy of BRWD1 for trisomy 21 contributing gene was mentioned in PMID 24740065 (Letourneau et al Nature vol 508 pg 349). Thus I suggest that the authors include this paper in the introduction somewhere in the paragraph starting on line 140.

I found the exploration of the role of BRWD1 of great importance in the Down syndrome field, and thus this MS provides an initial and serious step in this direction.

Response: We thank this Reviewer for their insightful and positive comments on our manuscript. We have cited the paper mentioned above, and please find our detailed responses to the concerns raised below.

Concerns/Criticisms:

1. The study is entirely done on the mouse model Ts65Dn. Although this model has been widely used, it also has disadvantages since it has a partial trisomy MMU16 AND a partial trisomy MMU17 that complicates the relevance to human trisomy 21.

Response: We agree with the Reviewer that the Ts65Dn mouse model, like most rodent models of complex human disease, does not fully recapitulate all aspects of human DS. Such caveats are now discussed further in the revised manuscript (see below and “Discussion” section). In addition, we have adjusted the title of the manuscript

accordingly to make it clear that these studies were conducted in a trisomic mouse model of DS.

“While no mouse model can fully recapitulate the complexity of the human DS disease state, the Ts65Dn line is unique amongst other commonly used DS models in that it carries triplicated MMU16 genes on a separate, freely segregating chromosome, *vs.* other models such as DP(16)1/Yey or TS1Cje that harbor the HSA21-orthologous triplication as an extension or replacement on existing chromosomes. Critically, this extra chromosome phenocopies the genomic instability caused by autosomal aneuploidy found in human DS, in addition to the HSA21 gene triplication. A recent study comparing all three DS mouse models found that Ts65Dn exhibited features that most closely resembled the human symptomatic arc – particularly during embryonic time points, when Ts65Dn shows alterations in neuroanatomical features, cytoarchitecture (e.g. neocortical expansion/neurogenesis) and aberrant gene expression – while neither DP(16)1/Yey or TS1Cje exhibited major prenatal symptoms. Furthermore, in adults, Ts65Dn mice were found to display the most altered behavioral responses in hippocampal-dependent tasks and gene expression⁴¹. Interestingly, Ts65Dn males exhibited more profound developmental deficits and behavioral changes in adults compared with females, which paralleled aspects of sex differences observed in DS patients, as well as the sex differences observed in the current study. One notable drawback of the Ts65Dn model is that the additional MMU16 chromosome also contains the centromeric region of MMU17, including ~30 non-HSA21 orthologous PCGs. This complication of the Ts65Dn model highlights the importance of our experimental design utilizing a specific gene rescue approach that maintains the Ts65Dn trisomic genetic background in order to more precisely determine if non-HSA21 genes are driving the deficits observed in these mice.”

2. Mouse Ts65Dn also contains 3 copies of the DYRK1A gene that several studies have shown its importance in the hippocampal FTP and other functional tests. Thus one needs to clarify/explain/discuss why the Ts65Dn with 3x Dyrk1a and 2x Brwd1 is "normal".

Response: This is certainly a very interesting point being raised by the Reviewer, and we now provide additional discussion on this idea in the “Discussion” section of the manuscript (see below).

“Additionally, it is important to consider that BRWD1 is not the only epigenetic regulator encoded on HSA21. Elevated expression of another HSA21-encoded gene, *DYRK1A*, has been observed in rodent models of DS and has been implicated in DS phenotypes. Like BRWD1, *DYRK1A* overexpression can impair cognition in mice, and restoring *Dyrk1a* to euploid copy number (or pharmacologically inhibiting it) can rescue DS-related cognitive impairments, skeletal abnormalities and Alzheimer’s disease related phenotypes in trisomic mice. Since BRWD1 and *DYRK1A* overexpression have strikingly similar effects on cognition, we hypothesize that they may function in the same pathway. *DYRK1A* also been shown to participate in BAF activity, most likely via its kinase activity. In support of this possibility, a study of phosphoproteins that showed treatment with a *DYRK1A* kinase inhibitor in Ts65Dn brain rescued altered

phosphorylation of BAF complex subunits. Importantly, our data suggests that BRWD1 may also contribute to BAF function through its binding to histone modification marks, thereby guiding the genomic targeting of BAF to facilitate chromatin restructuring. Therefore, our findings support a model in which BRWD1 renormalization may rescue the effects of BAF dysregulation, despite high levels of DYRK1A. How *BRWD1* may genetically interact with *DYRK1A* or other HSA21-encoded proteins to regulate cognition is an important avenue for future research. In future studies aimed at validating and investigating DS-related mechanisms in models for human brain development, such as cerebral organoids, it will be important to evaluate the effects of single allele mutations for key HSA21 genes (e.g., *BRWD1*) in DS patient-derived systems and examine DS-related molecular signatures for evidence of genetic contributions and interactions in DS.”

3. Since Ts65Dn is not the perfect model for trisomy 21, and many claims (including therapeutic ones) based on this model did not correspond to what is seen in humans, I suggest the authors to also perform some additional experiments in human trisomy 21 cells. For example inactivate one BRWD1 allele in a trisomy cell line by allelic CRISPR for example, and look at the differences in the transcriptome, or differences in neuronal cells after differentiation. Without a validation in a human cellular system, this study will be "yet one more Ts65Dn"... In contrast, the human experiment will make this study an important contribution to the DS research.

Response: While we completely agree with this Reviewer that no mouse model of DS perfectly recapitulates Trisomy 21 in humans (see discussion above), the experiment proposed involving hiPSC neurons from DS subjects $-/+$ genetic rescue using allele specific CRISPR-based approaches is, in our opinion, well beyond the scope of the current manuscript. Such an experiment would require recruitment of new patients, generation of novel iPSC lines, allele specific genetic manipulations using CRISPR (which are not trivial), neuronal differentiation and then downstream analyses – while interesting, such experiments would most definitely require additional collaborators, would be unnecessarily expensive and may take years to complete. Therefore, we feel that such explorations (now mentioned in the Discussion) should be saved for inclusion in future manuscripts.

Having said this, since the overarching goal of this Reviewer’s criticism, as we read it, is to better understand the relevance of our rodent findings to the human condition, we fully agree that providing additional evidence of such relationships are important for the current study. As such, we performed multiple bioinformatics analyses comparing gene expression profiles in euploid vs. Ts65Dn vs. Ts65Dn;*Brwd1*^{+/-} brain to human DS snRNA-seq data, results which are now included in the revised manuscript and are outlined below.

First, in an attempt to confirm the etiological relevance of the Ts65Dn mouse model to human DS at the level of transcription, we performed RNA-seq profiling on E17.5 Ts65Dn vs. euploid forebrain tissues. Differential expression analyses demonstrated that the Ts65Dn mouse model exhibits robust transcriptional changes that overlap

significantly with human DS single-nuclei gene expression profiles in postmortem PFC (Extended Data Fig. 3a-b). Of note, we found that only upregulated genes in Ts65Dn vs. euploid mice overlapped significantly with human DS associated gene expression, which included genes enriched for pathways associated with cellular development, neuronal differentiation and synaptic transmission. These data suggest that inappropriate induction of transcripts related to neuronal function may contribute to DS-related phenotypes in Ts65Dn mice (Extended Data Fig. 3c-d).

Next, to further evaluate the effects of *Brwd1* rescue within the context of adult brain, we performed RNA-seq on hippocampal tissues from 6-week-old male animals, comparing Ts65Dn vs. euploid and Ts65Dn;*Brwd1*^{+/-} genotypes (Fig. 2a). Pairwise comparisons identified 963 DE genes (FDR < .1) between Ts65Dn vs. euploid, and 801 DE genes (FDR < .1) between Ts65Dn;*Brwd1*^{+/-} vs. Ts65Dn, with ~17% of DE genes in Ts65Dn vs. euploid mice being significantly reversed in their expression with *Brwd1* copy number restoration (Fig. 2b). Consistent with our earlier analyses examining E17.5 forebrain tissues (Extended Data Fig. 3), odds-ratio assessments revealed significant overlaps with human DS associated gene expression, specifically for PCGs upregulated in Ts65Dn vs. euploid, and for those downregulated in Ts65Dn;*Brwd1*^{+/-} vs. Ts65Dn animals, indicating that *Brwd1* copy number restoration significantly reverses trisomic gene expression patterns in Ts65Dn male hippocampus that are relevant to human DS (Fig. 2c). Functional annotation analysis of rescued PCGs in adult male hippocampus against GO databases demonstrated significant enrichment of gene sets related to neuronal differentiation, neuronal morphology and synaptic function, consistent with observed deficits in synaptic function and hippocampal memory in male Ts65Dn mice (Fig. 2d).

Our data clearly demonstrate that the Ts65Dn mouse model robustly recapitulates many of the aberrant gene expression profiles observed in human DS brain (particularly in the case of transcriptionally upregulated genes), and our results further indicate that most of the loci displaying rescued expression following *Brwd1* copy number restoration in Ts65Dn mice do indeed significantly overlap with dysregulated genes identified in the human condition. In sum, we remain confident that our findings in mice have translational relevance in the context of human DS.

4. I found the model on extended data figure 10 rather weak; why for example the protein loaded with the 3 yellow circles (BAF complex) works in euploid brain and not in the Ts65Dn (top and bottom of the scheme)?

Response: We greatly apologize for any confusion regarding our Extended Data model Figure. We have chosen to remove this model from the manuscript, as it does not fully encapsulate the main points of the manuscript. We have, however, greatly extended the Discussion section of the manuscript to better highlight the findings presented in this paper.

5. Yansheng Liu in PMID: 29089484 (NCOMM 2017) has shown that proteins in complexes are well buffered in trisomy 21 and their amount is not increased as one might

expect from the RNA data. Wondering if the authors could measure the BRWD1 protein in the mouse hippocampal extracts.

Response: This is an excellent point. However, as mentioned in the manuscript, currently available antibodies for BRWD1 are non-specific (and we believe that we have tried every available antibody on the market) and may not provide an accurate depiction of *Brwd1* expression in mouse brain lysates (hence the reason that we generated a novel BRWD1 tagged mouse line for studies examining BAF complex interactions). Given our robust data indicating that genetic rescue of *Brwd1* overexpression in these animals rescues (in some cases, fully rescues) synaptic, transcriptional and behavioral deficits observed in trisomic animals, it is difficult to imagine a scenario in which dysregulation of BRWD1 protein expression itself would not be affected in our different comparisons. In future studies, we will continue to pilot new antibodies that may become available, or may even consider developing our own antibody, time and resources permitting.

REVIEWERS' COMMENTS

Reviewer #1 (Remarks to the Author):

The authors have addressed all reviewer concerns and done a heroic effort to make this manuscript both accessible to readers from both the clinical, basic neuroscience and epigenomics community. The new experiments, analysis and discussion of sex differences make a clear and convincing argument for Brwd1 and BAF in Down Syndrome. The revised manuscript was a joy to read and will have a significant impact on the field.

Two minor comments

1. For Figure 2A and 2B as well as Extended Figure 7a and b and Extended Figure 8a and b, the measure is row z-score, but if you want to compare across rows (the different genetic conditions, which I think is what you want to emphasize here) then it should be column z-scores to allow for a comparison of expression levels of each gene (column) across conditions (rows).
2. Please verify that all transcriptomic and epigenomic datasets will be made available on NCBI GEO upon publication.

Reviewer #2 (Remarks to the Author):

This is an interesting paper that advances molecular understanding of neuronal gene expression changes that contribute to altered brain function in Down Syndrome. In my opinion the authors have done an excellent job responding to the concerns of the reviewers. In particular the contextualization of the sex differences is much stronger now. I think the paper will be of interest to a broad neurobiology readership.

Reviewer #3 (Remarks to the Author):

The authors have provided extensive and well thought answers to my criticisms and comments. I strongly suggest to specify in the title the Ts65Dn mouse used. The new title could be: "Rescue of deficits by Brwd1 copy number restoration in the Ts65Dn mouse model of Down syndrome"

The problem of the Dyrk1A remains, and the authors dealt with this in the discussion. The potential link between Dyrk1a and Brwd1 needs to be studied in a subsequent paper, since it has implications in potential treatment options with Dyrk1a-specific inhibitors. Both genes (Dyrk1a and Brwd1) have a PLI of 1 that strongly indicates that they are both dosage sensitive, and are therefore excellent candidates for some DS phenotypes.

The protein levels of Brwd1 could be measured with a proteome analysis in mouse brain lysates of specific brain regions.

Response to Referees

Reviewer #1:

The authors have addressed all reviewer concerns and done a heroic effort to make this manuscript both accessible to readers from both the clinical, basic neuroscience and epigenomics community. The new experiments, analysis and discussion of sex differences make a clear and convincing argument for Brwd1 and BAF in Down Syndrome. The revised manuscript was a joy to read and will have a significant impact on the field.

Response: We very much thank the Reviewer for their positive and helpful feedback on our manuscript, and for agreeing that our paper is now suitable for publication at *Nature Communications*.

Two minor comments

1. For Figure 2A and 2B as well as Extended Figure 7a and b and Extended Figure 8a and b, the measure is row z-score, but if you want to compare across rows (the different genetic conditions, which I think is what you want to emphasize here) then it should be column z-scores to allow for a comparison of expression levels of each gene (column) across conditions (rows).

Response: We apologize for any confusion regarding this point. Since we flipped the orientation of these heatmaps for visual presentation purposes, the Reviewer is correct that labeling them as “Row Z score” is confusing. As such, we have re-labelled them to read as “Gene Z score,” which more accurately reflects the comparisons that we are making.

2. Please verify that all transcriptomic and epigenomic datasets will be made available on NCBI GEO upon publication.

Response: All genomics datasets have been uploaded to GEO and have been publicly released to the field. This information can be found in the “Data Availability” section of the manuscript.

Reviewer #2

This is an interesting paper that advances molecular understanding of neuronal gene expression changes that contribute to altered brain function in Down Syndrome. In my opinion the authors have done an excellent job responding to the concerns of the reviewers. In particular the contextualization of the sex differences is much stronger now. I think the paper will be of interest to a broad neurobiology readership.

Response: We greatly appreciate this Reviewer's continued enthusiasm regarding our manuscript and their feeling that our findings "will be of interest to a broad neurobiology readership."

Reviewer #3

The authors have provided extensive and well thought answers to my criticisms and comments.

Response: We thank the Reviewer for their constructive suggestions throughout the review process, which have greatly helped to improve both the quality and scope of our manuscript.

I strongly suggest to specify in the title the Ts65Dn mouse used. The new title could be:

"Rescue of deficits by Brwd1 copy number restoration in the Ts65Dn mouse model of Down syndrome"

Response: We have re-titled the manuscript in accordance with the Reviewer's suggestion.

The problem of the Dyrk1A remains, and the authors dealt with this in the discussion. The potential link between Dyrk1a and Brwd1 needs to be studied in a subsequent paper, since it has implications in potential treatment options with Dyrk1a-specific inhibitors. Both genes (Dyrk1a and Brwd1) have a PLI of 1 that strongly indicates that they are both dosage sensitive, and are therefore excellent candidates for some DS phenotypes.

Response: We fully agree that future studies aimed at investigating potential interactions between Dyrk1a and Brwd1 in the context of trisomy will be needed to elucidate their overlapping and discrete actions in the precipitation of disease. We look forward to exploring these interactions further in subsequent studies.

The protein levels of Brwd1 could be measured with a proteome analysis in mouse brain lysates of specific brain regions.

Response: While we agree that proteomic analyses could possibly be used to further explore Brwd1 regulation in the trisomic brain in the absence of suitable antibodies – experiments that we plan to perform in subsequent studies – such experiments will require the generation and optimization of spike-in peptides for accurate LC-MS/MS quantifications. Additionally, since very little remains known regarding potential posttranslational modifications occurring on Brwd1 in brain, such modifications would also need to be explored in order to ensure accurate quantification of Brwd1 levels in tissue extracts using LC-MS/MS. Given the potential time required to optimize these proteomics approaches, as well as our robust findings indicating that Brwd1 copy number

restoration rescues molecular, physiological and behavioral deficits observed in trisomic mouse brain (effects that most certainly require restoration of aberrant Brwd1 proteins levels), we feel that such studies should be reserved for future publications.